# 🌹ROSE-TTA: RELIABLE ONLINE STRUCTURAL ENHANCEMENT FOR TEST-TIME ADAPTATION

## ABSTRACT

Large-scale vision-language models like CLIP exhibit remarkable zero-shot generalization but suffer significant performance degradation under real-world distribution shifts. Although recent cache-based test-time adaptation (TTA) methods mitigate the issues, they are limited by: (i) unreliability in cache construction, as entropy-based sample selection is insufficient under distribution shifts; and (ii) incomplete cache information at inference, with both imbalanced category information caused by sequential online updates and insufficient sample-specific information for next online instance. To address these limitations, we propose **ROSE-TTA** (Reliable Online Structural Enhancement for Test-Time Adaptation), a unified framework that enhances both cache construction and utilization for more reliable and stable adaptation. For construction, we introduce a noise-aware uncertainty measure that combines entropy with perturbation-based prediction stability to robustly select cache entries. To complete the cache information for utilization, we develop a graph-based structural completion strategy, which effectively mitigates class imbalance and completes global information by transferring information between text embeddings and cached features. Additionally, we introduce a sample-specific refinement mechanism to dynamically update cache features and incorporate local information of each online test sample. Experiments on 15 widely used datasets demonstrate the effectiveness of our method.

## 1 INTRODUCTION

The emergence of large-scale vision-language models (VLMs), such as CLIP (Radford et al., 2021), has profoundly reshaped the landscape of multi-modal learning. By effectively aligning visual and textual modalities through contrastive pretraining on web-scale datasets, these models demonstrate remarkable zero-shot transfer capabilities across diverse downstream tasks. However, deploying CLIP in real world remains challenging, especially on specific data with distribution shifts between the test and pretrained distributions (Shu et al., 2022; Han et al., 2024; Karmanov et al., 2024).

A common approach to address distribution shift at test time is test-time adaptation (TTA) (Sun et al., 2020; Wang et al., 2021; Chen et al., 2022), which has also been leveraged to improve CLIP's zero-shot capability (Shu et al., 2022; Karmanov et al., 2024). Test-time prompt tuning (TPT) (Shu et al., 2022; Feng et al., 2023; Karmanov et al., 2023; Yoon et al., 2024) fine-tunes textual prompts at inference, often guided by entropy minimization. Despite their effectiveness in addressing distribution shifts, these methods are computationally expensive due to backpropagation. To improve efficiency, cache-based TTA methods have emerged (Karmanov et al., 2024; Han et al., 2024). These approaches introduce a dynamic cache to store representative test features online, which are used to refine predictions without backpropagation, providing a lightweight online adaptation mechanism.

Despite their computational efficiency, existing cache-based TTA methods suffer from two fundamental limitations that compromise their robustness under distribution shifts. First, cache construction lacks reliability. Relying solely on entropy-based selection fails to effectively eliminate noisy or misclassified instances (Nguyen et al., 2023a; Han et al., 2024; Shamsi et al., 2024; Zhou et al., 2025), leading to corrupted cache. Second, cache information available during inference is inherently incomplete. At global level, sequential online updates induce class imbalance within cached features. Especially during the early adaptation stage, this random arrival leads to unequal cache capacities across classes. This imbalance causes the cache to favor the majority classes and provides insufficient

guidance for the minority classes, ultimately biasing model predictions and even leading to error accumulation during continuous adaptation (Zhang et al., 2023). Additionally, cached features lack local information of each incoming sample, providing limited sample-specific guidance for the online instance. These issues hinder both the stability and generalization of cache-based adaptation. However, prior work (Karmanov et al., 2024; Han et al., 2024; Zhou et al., 2025) primarily focuses on either construction or utilization, treating them as separate problems. We recognize these are inherently coupled: unreliable cache construction cascades into biased utilization, while incomplete utilization fails to leverage even high-quality cached samples.

To systematically address these limitations, we propose **ROSE-TTA** (**R**eliable **O**nline **S**tructural **E**nhancement for **T**est-**T**ime **A**daptation), a novel and unified framework that enhances the reliability and the stability in cache update and integrates complementary global (class level) and local (sample-specific) information for robust cache-based adaptation. In cache construction, we introduce an improved cache update mechanism to synergistically combines entropy with a noise-enhanced uncertainty measure, which evaluates prediction robustness under perturbations as a complementary signal to confidence. This dual-criterion approach ensures that only the most reliable and consistently stable online instances are incorporated into the cache, improving the cache reliability. Moreover, to complement global semantic information in the cache, we propose a novel graph-based structural completion strategy (Li et al., 2024). The method reconstructs the categorical graph with the semantic relationships between classes from text embeddings and test specific information from cached features, mitigating class imbalance and strengthening the representation of underrepresented categories within the cache (Zhang et al., 2024a). We also design a sample-specific refinement mechanism that updates cached features on-the-fly using the information of each test sample, incorporating local information and improving alignment between the cache and instance.

We evaluate **ROSE-TTA** on 15 widely used datasets, covering typical evaluation scenarios such as domain generalization and cross-dataset. The experimental results demonstrate the effectiveness of our method on enhancing the reliability and adaptability of cache-based TTA.

## 2 REVISITING CACHE-BASED TEST-TIME ADAPTATION FOR CLIP

### 2.1 PRELIMINARY

**CLIP (Radford et al., 2021).** CLIP is known for the remarkable ability in vision-language representations learning through large-scale training in image-text data. The pretrained CLIP model consists of an image encoder $\mathcal{F}_{\theta_I}(\cdot)$ and a text encoder $\mathcal{F}_{\theta_T}(\cdot)$, with $\theta_I$ and $\theta_T$ denoting the model parameters, respectively. Based on a zero-shot $C$-class classification task, for each class $c \in \{1, \ldots, C\}$, we generate a text prompt $\boldsymbol{t}_c$ by instantiating a template such as "*a photo of a [class]*", where "*[class]*" is replaced with the name corresponding to class $c$. Each text prompt $\boldsymbol{t}_c$ is then encoded as $\boldsymbol{f}_c = \mathcal{F}_{\theta_T}(\boldsymbol{t}_c)$ and the image $\boldsymbol{x}$ is encoded as $\boldsymbol{f}_{\boldsymbol{x}} = \mathcal{F}_{\theta_I}(\boldsymbol{x})$. Collecting all text embeddings as the matrix $\boldsymbol{W}_C = [\boldsymbol{f}_1, \boldsymbol{f}_2, \ldots, \boldsymbol{f}_C]$, CLIP seeks to associate the image feature $\boldsymbol{f}_{\boldsymbol{x}}$ with the most semantically relevant text feature from $\boldsymbol{W}_C$. The probability of $\boldsymbol{x}$ to be classified as class c is $p(\hat{y} = c \mid \boldsymbol{x}) = \frac{\exp(\cos(\boldsymbol{f}_{\boldsymbol{x}}, \boldsymbol{f}_c)/\tau)}{\sum_{k=1}^{C} \exp(\cos(\boldsymbol{f}_{\boldsymbol{x}}, \boldsymbol{f}_k)/\tau)}$, where $\cos(\cdot, \cdot)$ denotes cosine similarity and $\tau$ is a temperature parameter. The most relevant class is obtained from CLIP by $\arg\max_c p(\hat{y} = c \mid \mathbf{x})$. For subsequent analysis, we denote the predicted probabilities $p(\hat{y} = c \mid \boldsymbol{x})$ over all $C$ classes as $P_{clip}$.

**Test-time adaption based on key-value cache for CLIP.** As a training-free solution that adapts pre-trained models to test data with distributional shift (Tahir et al., 2022; Zhang et al., 2024b; Gao et al., 2025), test-time adaptation adjusts model predictions on-the-fly to better align with the test data. A prominent family of methods leverages a *key–value cache* that accumulates reliable test samples to refine CLIP's predictions (Han et al., 2024; Karmanov et al., 2024).

In the cache-based methods, a memory $(\boldsymbol{F}, \hat{\boldsymbol{L}})$ is introduced to store N historical features $\boldsymbol{F} \in \mathbb{R}^{N \times d}$ and their corresponding (pseudo-)labels $\hat{\boldsymbol{L}} \in \mathbb{R}^{N \times C}$. The interaction between a new feature $\boldsymbol{f}_{test}$ and the cache follows a unified paradigm:

$$P_{cache}(\boldsymbol{f}_{test}) = A(\boldsymbol{f}_{test}\boldsymbol{F}^T) \, \hat{\boldsymbol{L}}, \tag{1}$$

where $\boldsymbol{f}_{test}\boldsymbol{F}^T$ denotes the affinity scores (Karmanov et al., 2024) between the new feature and cached features, and $A(\cdot)$ is an activation function that maps these affinities into weights. Finally, the

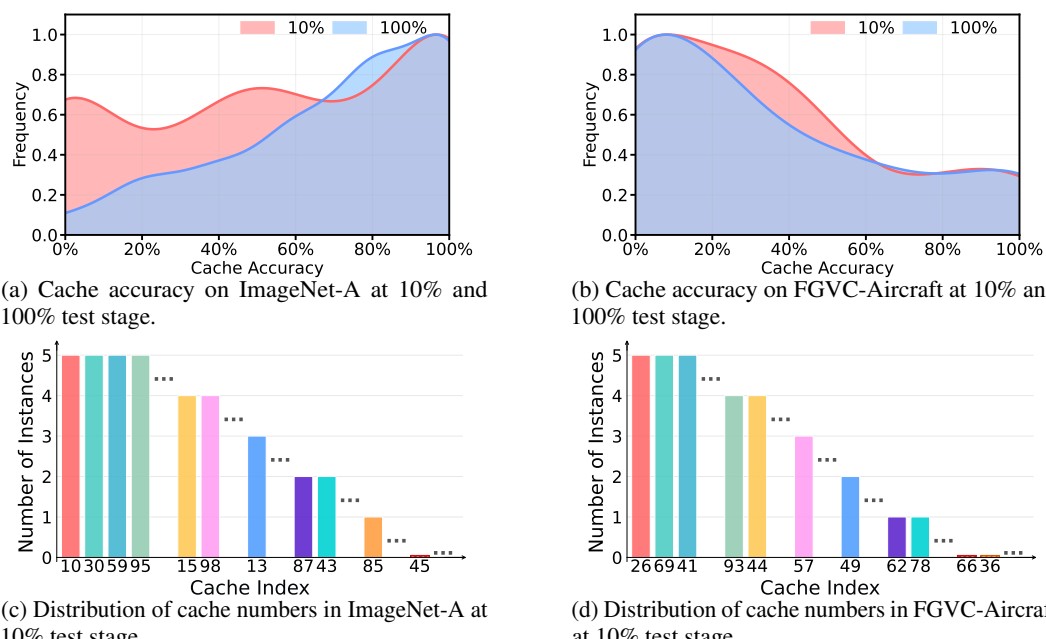

(a) Cache accuracy on ImageNet-A at 10% and 100% test stage.

(b) Cache accuracy on FGVC-Aircraft at 10% and 100% test stage.

(c) Distribution of cache numbers in ImageNet-A at 10% test stage.

(d) Distribution of cache numbers in FGVC-Aircraft at 10% test stage.

Figure 1: **Category and accuracy statistics of the entropy-based online cache.** The online updated cache can be unreliable (a and b) and imbalanced (c and d).

complete prediction combines the original CLIP output with the cache contribution:

$$P_{final} = P_{clip} + \alpha \cdot P_{cache}, \tag{2}$$

where $\alpha$ is a scaling factor to balance the influence of cache information. This formulation provides a standardized cache-based mechanism for refining the CLIP zero-shot predictions, where the effectiveness depends on how $(\boldsymbol{F}, \hat{\boldsymbol{L}})$ are updated during inference.

## 2.2 ISSUES OF CACHE-BASED TEST-TIME ADAPTATION

Although the cache-based methods achieve efficient test-time adaptation for zero-shot classification of CLIP (Wang et al., 2021; Zhang et al., 2024b), the entropy-based cache construction and utilization are not always stable and reliable (Han et al., 2024; Zhou et al., 2025). Samples with low entropy can still be misclassified in unseen test data distributions (Lee et al., 2024). In Figures 1a and 1b, we report the precision of the features stored in the entropy-based online cache (Karmanov et al., 2024) at the beginning and end of the online test stage on two different datasets. We found that even when selecting features with minimum entropy, many of them are cached under incorrect labels, especially for unfamiliar datasets (Figure 1b). The problem is more severe at the earlier test stage (10%). These findings indicate that entropy alone is an insufficient criterion for cache construction and update.

Moreover, since online test samples typically arrive in random sequential order in practice, the cache update mechanism inherently induces class imbalance, particularly at the early adaptation stage. We counted the number of per-class features in the online updated cache again for different datasets after processing 10% of the stream. As shown in Figures 1c and 1d, the number of cache features are extremely imbalanced among categories in both cases. During the early adaptation stage, this random arrival leads to unequal cache capacities across classes. Some classes may have accumulated 5 cached samples (full capacity), while others have less (1 or 2), or even no samples. The class imbalance biases the predictions toward head classes with larger caches while neglecting classes with few or no entries. Consequently, the model's predictions are heavily skewed toward classes with richer cache representations, yielding skewed and inaccurate predictions. The inaccuracies can even accumulate during online learning, leading to progressively worse overall outcomes.

Additionally, the incoming instance in the online test stream is unpredictable, which can be a new class or a an unseen style. Although the cache is updated online, there remains a lack of local instance information of such specific test samples, leading to unstable predictions.

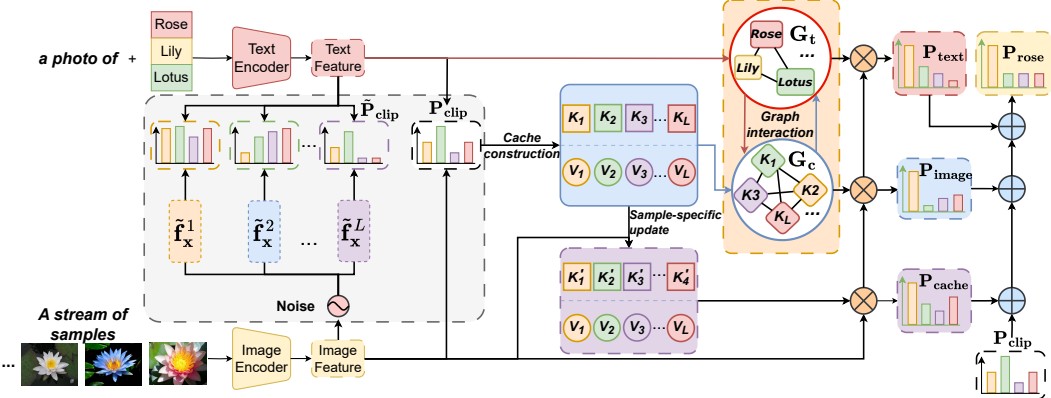

Figure 2: **Illustration of Rose-TTA.** We propose uncertainty aware cache construction to store more reliable test samples in the cache. During utilization, the cache is further enhanced by graph-based structural completion and sample-specific refinement, which inject the global category information and local instance information, respectively.

Therefore, in this paper, we propose an enhanced cache for online test-time adaptation by uncertainty-aware cache construction and graph-based information completion.

## 3 METHODOLOGY

To reduce the issues of unreliable cache construction and biased predictions caused by incomplete information, we propose **ROSE-TTA**, a more reliable and stable cache for CLIP test-time adaptation, consisting of uncertainty-aware cache construction and graph-based information completion with sample-specific refinement for cache utilization. An illustration of our method is shown in Figure 2.

### 3.1 UNCERTAINTY-AWARE CACHE CONSTRUCTION

To directly combat the unreliability of cache construction, where sole reliance on entropy often leads to noisy or misclassified samples (Nguyen et al., 2023b; Han et al., 2024; Lee et al., 2024), we introduce an improved cache update mechanism. This mechanism ensures that only the most reliable and consistently stable features are incorporated into the cache, improving the quality of the cache construction features.

**Noise-enhanced uncertainty estimation.** The noise perturbation mechanism serves a critical role in assessing prediction reliability under distribution shifts. While entropy-based selection can identify confident predictions, it fails to distinguish between genuinely stable samples and overconfident but fragile predictions. To improve the reliability of samples stored in the cache, we propose a noise-enhanced uncertainty estimation mechanism that evaluates the stability of predictions under controlled perturbations. The mechanism is used to select the stable features for cache construction and update during online test-time adaption.

Given an input test sample $\boldsymbol{x}$, we first obtain its CLIP prediction $\hat{c} = \arg\max_c P_{clip}$ to find the corresponding class-wise cache. To access the prediction stability of this sample, i.e., the prediction consistency under small perturbations, we generate $n$ augmented features by adding calibrated Gaussian noise on the original feature:

$$\tilde{\boldsymbol{f}}_{\boldsymbol{x}}^i = \boldsymbol{f}_{\boldsymbol{x}} + \epsilon_i \cdot \sigma, \quad \epsilon_i \sim \mathcal{N}(0, \boldsymbol{I}), \quad i = 1, \ldots, n, \tag{3}$$

where $\sigma$ controls the noise magnitude and $n$ is the number of augmentation, which is used to balance perturbation strength and feature semantic preservation. With the predictions on the noise-perturbed features, we obtain the prediction stability for the sample $\boldsymbol{x}$ by:

$$\mathbf{d}(\boldsymbol{x}) = \frac{1}{n} \sum_{i=1}^n |p(\hat{y} = \hat{c} \mid \tilde{\boldsymbol{f}}_{\boldsymbol{x}}^i) - p(\hat{y} = \hat{c} \mid \boldsymbol{f}_{\boldsymbol{x}})|, \tag{4}$$

where $p(\hat{y} = \hat{c} \mid \tilde{\boldsymbol{f}}_{\boldsymbol{x}}^i)$ and $p(\hat{y} = \hat{c} \mid \boldsymbol{f}_{\boldsymbol{x}})$ denote the predicted probability on class $\hat{c}$ given the noise-perturbed feature $\tilde{\boldsymbol{f}}_{\boldsymbol{x}}^i$ and original feature $\boldsymbol{f}_{\boldsymbol{x}}$, respectively. The prediction stability metric

measures how confidence changes under noise perturbations. Smaller $\mathbf{d}(\boldsymbol{x})$ indicates that the instance is more robust to pertubations and therefore more reliable for caching.

**Doubly robust cache.** To achieve a more reliable cache during online test-time adaptation, we adopt both entropy and noise-enhanced stability as our overall selection metric to construct a doubly robust cache $\mathcal{C} = (\boldsymbol{F}, \hat{\boldsymbol{L}}) = \{\mathcal{C}_c\}_{c=1}^C = \{(\boldsymbol{F}_c, \hat{\boldsymbol{L}}_c)\}_{c=1}^C$. Following TDA (Karmanov et al., 2024), $\boldsymbol{F}_c, \hat{\boldsymbol{L}}_c$ are the raw features of the historical samples and the one-hot labels of class $c$, respectively. When the cache $\mathcal{C}_c$ for class $c$ has not reached its maximum capacity, the new sample is directly appended along with its entropy and stability measure. Since the cache size cannot be infinite, when the cache reaches its capacity $n_{\mathcal{C}}$, the new sample is admitted to the cache $\mathcal{C}_c$ only if it surpasses existing cached samples on both metrics. Specifically, the replacement occurs if and only if both conditions are satisfied:

$$\begin{cases} \mathcal{H}(p(y|\boldsymbol{x})) < \mathcal{H}(p(y|\boldsymbol{x}_{j^*})) & \text{(lower entropy)} \\ \mathbf{d}(\boldsymbol{x}) < \mathbf{d}(\boldsymbol{x}_{j^*}) & \text{(higher stability)}, \end{cases} \tag{5}$$

where $\mathcal{H}$ and $\mathbf{d}$ are the entropy and prediction stability of the input sample. $j^*$ denotes the index of the weakest cached sample with the highest combined entropy and the lowest stability score. This dual-criterion approach ensures that the cache maintains samples that are not only confident(low entropy) but also robust to input perturbations, avoiding overconfidence under distribution shifts and improving the reliability of cached features for test-time adaptation.

## 3.2 GLOBAL AND LOCAL COMPLETION IN CACHE UTILIZATION

Beyond improving cache reliability during construction, our method also strengthens the utilization of the test-time cache by completing both global and local information.

**Graph-based structural completion.** As shown in Figure 1, since the test data sequence is random and unpredictable, the online test-time cache exhibit class imbalance and incomplete global task information, especially in the early test stage. This biases the predictions towards majority classes and neglects minority ones. For example, if no sample from class $c$ has appeared, the cache cannot provide sufficient corresponding categorical information for prediction. Consequently, when a new sample from $c$ arrives, the cache guidance tends to be noisy or meaningless, and even degrade the final prediction.

To mitigate class imbalance and complete the global information in online cache at inference, we propose a novel graph-based structural completion strategy. The method reconstructs the categorical graph of the task by integrating class information from both text embeddings and cache features. The graph addresses the class imbalance issue by introducing the global class information from text embeddings of all categories, while preserving the test distributional information by considering the cache features. Specifically, we construct three complementary graphs to capture different aspects of the information:

*1) Cache-graph* $\boldsymbol{G}_c \in \mathbb{R}^{N \times N}$: $\boldsymbol{G}_c = (\boldsymbol{F}^T \boldsymbol{F})$ models the relationships between cached features, where $\boldsymbol{F}$ containing all cached features. $\boldsymbol{G}_c$ provides the information of the test data in both class-level and instance-level, preserving the test specific information stored in the cache. Moreover, since cached features vary in reliability, we further introduce a reliability-weighted cache graph. First, we assign each cache sample a reliability weight $w_j = \sqrt{(1 - \mathbf{d}_j)/\mathcal{H}_j}$, where $\mathcal{H}_j$ and $\mathbf{d}_j$ are the entropy and stability scores of the $j$-th cached sample. Samples with lower entropy and higher stability have a higher $w_j$, indicating higher reliability and thus greater influence in the graph. The reliability weight matrix is then constructed as $\boldsymbol{W}_{jk} = \sqrt{w_j \cdot w_k}$ and used to refine the Cache-Graph by $\hat{\boldsymbol{G}}_c = (\boldsymbol{F}^T \boldsymbol{F} \odot \boldsymbol{W})$. Here $\odot$ denotes element-wise multiplication.

*2) Text-graph* $\boldsymbol{G}_t \in \mathbb{R}^{C \times C}$: $\boldsymbol{G}_t = (\boldsymbol{W}_C^T \boldsymbol{W}_C)$ encodes the semantic relationships of different classes at the text level, where $\boldsymbol{W}_C$ denotes the matrix of text embeddings for all classes. By considering all classnames, $\boldsymbol{G}_t$ provides the global category information of the test task. We reconstruct the imbalanced cache graph to provide complementary information for underrepresented classes. Specifically, the textual graph leverages semantic relationships between classes to enhance the logits of categories with insufficient cache samples, thereby producing more balanced predictions.

*3) Gate-graph* $\boldsymbol{M} \in \{0,1\}^{C \times N}$ is a binary mapping matrix that links cached samples to their corresponding classes. $\boldsymbol{M}_{cj}$ is set to 1 if the cached sample $\boldsymbol{x}_j$ belongs to class $c$, else 0.

We reconstruct the categorical graph by integrating the three graphs to incorporate both global class relations from text embeddings and cache-specific relations from cached samples:

$$\hat{\boldsymbol{G}} = \text{softmax}(\boldsymbol{G}_t \boldsymbol{M} \hat{\boldsymbol{G}}_c \boldsymbol{M}^T \boldsymbol{G}_t^T). \tag{6}$$

With the reconstructed graph $\hat{\boldsymbol{G}} \in \mathbb{R}^{C \times C}$, we derive two graph-enhanced prediction terms from the textual and visual paths for a test feature $\boldsymbol{f}_x$:

$$P_{\text{t}} = \boldsymbol{f}_{\boldsymbol{x}}(\hat{\boldsymbol{G}} \boldsymbol{W}_C^T)^T, \qquad P_{\text{i}} = \boldsymbol{f}_{\boldsymbol{x}}(\hat{\boldsymbol{G}} \boldsymbol{M} \boldsymbol{F}^T)^T. \tag{7}$$

This dual pathway expends the textual prediction to $P_t$ by injecting test-specific information stored in cache. Moreover, the cache prediction $P_i$ is enriched by the global class information from text embeddings, mitigating the class imbalance problem and strengthening the underrepresented categories in the cache.

**Sample-specific completion.** Except for the global class information, the cache can also lack local instance information since the next online test sample is usually unpredictable in real applications. If a specific test sample is different from the cached ones during online learning, the cache prediction can still be unreliable, limiting its adaptability. To incorporate local instance information in cache utilization, we propose a sample-specific cache refinement mechanism to dynamically update cached features in utilization based onincoming test samples.

Gradient-based TTA (Shu et al., 2022; Zhang et al., 2022a) usually refines the model parameters for each test sample by gradient backpropagation of entropy minimization, which, however, is computationally expensive. To achieve efficient sample-specific cache refinement, we propose a backpropogation-free method to directly infer the pseudo gradient according to the sample feature and its entropy value.

The basic refinement of the cached features for the test sample $\boldsymbol{x}$ is the feature $\boldsymbol{f}_x$. To control the update amount of each cache feature according to the cache prediction, we first obtain the averaged cache prediction $P_n = softmax\left(\frac{1}{n}\sum_{i=1}^{n} P_{cache}(\tilde{\boldsymbol{f}}_{\boldsymbol{x}}^i)\right)$ based on the noise-augmented features in Eq. (3). We then introduce two intensity control factors $\gamma = 1 - \mathcal{H}(P_n)$ and $\zeta = P_n - 1/C$ based on the averaged cache prediction, where $\mathcal{H}(P_n)$ denotes the normalized entropy of $P_n$. Here $\gamma$ is used to control the update intensity (more confidence, more change). $\zeta$ control the update direction, pushing up higher probability than uniform and pushing down lower probability than uniform, which is what entropy minimization tends to do. To compare the pseudo-gradient and actual gradient of entropy minimization, we provide more theoretical and empirical analyses in Appendix A.

Based on the text feature and control factors, we perform sample-specific refinement of the cache features as:

$$\hat{\boldsymbol{F}} = \boldsymbol{F} + \eta \cdot \gamma \cdot \zeta \cdot \boldsymbol{f}_{\boldsymbol{x}}, \tag{8}$$

where $\eta$ is a pseudo-learning rate, $\gamma$ and $\zeta$ represent the uncertainty and confidence of the averaged prediction, controlling the update intensity and direction. Following Eq. (1), The refined cache prediction is calculated by:

$$\hat{P}_{cache} = A\left(\boldsymbol{f}_{\boldsymbol{x}} \hat{\boldsymbol{F}}^T\right) \hat{\boldsymbol{L}}, \tag{9}$$

where $A(\cdot)$ is an activation function and $\hat{\boldsymbol{L}}$ denotes the one-hot pseudo labels of the cache features. This sample-specific refinement enables the cache to consider local instance information for adaptation efficiently, without gradient computation and backpropogation.

Overall, our method integrates multiple complementary sources of information to produce robust test-time predictions. The final prediction is calculated by:

$$P_{rose} = P_{clip} + \alpha \cdot (\hat{P}_{cache} + P_t + P_i), \tag{10}$$

where $\alpha$ is hyperparameter to control the magnitude of logits.

## 4 RELATED WORK

**Vision-language models.** The emergence of large-scale vision-language models has fundamentally transformed the landscape of multi-modal understanding. Early pioneering work CLIP (Radford

et al., 2021) demonstrated that contrastive learning on web-scale image-text pairs enables powerful zero-shot transfer capabilities across diverse visual recognition tasks. Building upon this foundation, ALIGN(Jia et al., 2021) demonstrated that scale matters significantly and FILIP(Yao et al., 2022) introduced fine-grained cross-modal alignment through token-level interactions, moving beyond global image-text matching. With ongoing research, works such as InternVL3.5 (Wang et al., 2025) further advance open-source multimodal models in versatility. Concurrently, studies like Qwen-Image (Wu et al., 2025) focus on image generation and DINOv3 (Siméoni et al., 2025) continues to push the boundaries of self-supervised learning.

**Test-time adaptation for vision-language models.** Test-time adaptation (TTA) (Sun et al., 2020; Nado et al., 2020; Wang et al., 2021) emerged to address distribution shift between training and test environments without requiring labeled target data, and has recently been extended to vision-language models with their dual-modal structure and rich semantics. Early works explored *gradient-based prompt tuning*, such as TPT (Shu et al., 2022),C -TPT (Yoon et al., 2024) and DynaPrompt(Xiao et al., 2025), which adapt text prompts via entropy minimization, richer augmentations, or calibration objectives. While effective, these methods incur high computational overhead and may suffer from instability. To improve efficiency, *training-free approaches* were proposed. Cache-based methods (e.g., Tip-Adapter (Zhang et al., 2022b), HisTPT (Tang et al., 2023), TDA (Karmanov et al., 2024)) leverage stored representative samples or multi-granularity knowledge to refine predictions without gradient updates, while PromptAlign (Karmanov et al., 2023) instead mitigates distribution shifts by aligning test and source statistics. Motivated by the limitations of entropy-based objectives, DOTA (Han et al., 2024) and Bayesian TTA (Zhou et al., 2025) provide principled alternatives through distribution estimation and uncertainty quantification respectively. Despite progress, current methods often over-rely on entropy-based selection, suffer from class imbalance where underrepresented classes receive insufficient cache guidance, and employ static feature representations that fail to adapt to evolving test distributions or utilize the semantic structure in text embeddings.

## 5 EXPERIMENTS

**Datasets.** We evaluate on two commonly used benchmarks in TTA on zero-shot CLIP (Shu et al., 2022; Karmanov et al., 2024; Han et al., 2024). The *cross dataset* benchmark covers ten heterogeneous datasets, including Aircraft (Maji et al., 2013), Cars (Krause et al., 2013), Pets (Parkhi et al., 2012), Flower102 (Nilsback & Zisserman, 2008), Food101 (Bossard et al., 2014), Caltech101 (Fei-Fei, 2004), SUN397 (Xiao et al., 2010), DTD (Cimpoi et al., 2014)), EuroSAT (Helber et al., 2019), and UCF101 (Soomro et al., 2012). The diverse datasets allows us to assess adaptability across distinct semantic and visual domains. The *out-of-distribution(OOD)* benchmark consists of ImageNet(Deng et al., 2009) and its four variants: ImageNet-A (Hendrycks et al., 2021b), ImageNet-V2 (Recht et al., 2019), ImageNet-R (Hendrycks et al., 2021a), and ImageNet-Sketch (Wang et al., 2019), which provide a rigorous test of robustness under different types of distribution shift.

**Implementation details.** Our experiments are conducted on the pre-trained CLIP model (Radford et al., 2021), which comprises an image encoder and a text encoder. Following (Karmanov et al., 2024), we adopt ResNet-50 (He et al., 2016) and ViT-B/16 (Dosovitskiy et al., 2020) as the image encoder backbones, and a Transformer (Vaswani et al., 2017) as the text encoder. To reflect real-world online test-time adaptation scenarios, we set the test batch size to 1. For hyperparameters, we fix $n$ / $\sigma$ / $\eta$ to 10 / 0.1 / 0.01 across all datasets. The cache capacity $n_{\mathcal{C}}$ is set to 5 for all dataset–backbone combinations. The coefficient $\alpha$ is tuned individually for each dataset to adapt to the specific scenario. We adopt the same hand-crafted prompt templates as in Karmanov et al. (2024). Top-1 accuracy is used as the evaluation metric. All experiments are performed on a single NVIDIA RTX 4090 GPU.

### 5.1 COMPARISONS

**Baselines.** In this section, we compare our method mainly with two kinds of methods: (1) test-time propmt tuning methods: TPT (Shu et al., 2022), C-TPT (Yoon et al., 2024), HisTPT (Tang et al., 2023), MTA (Zanella & Ben Ayed, 2024), and PromptAlign (Karmanov et al., 2023). (2) cache-based efficient test-time adaptation methods: TDA (Karmanov et al., 2024) and BCA (Zhou et al., 2025).

**Results on the cross-dataset benchmark.** We first compare ROSE-TTA with state-of-the-art methods on the *cross-dataset* benchmark (Table 1). For ResNet-50, ROSE-TTA achieves the best

Table 1: **Comparisons on the cross-dataset setting.** Our method outperforms the alternatives on five of the ten datasets and achieves best overall performance based on both ResNet-50 and ViT-B/16. The best and runner-up results are bolded and underlined, respectively.

| Method | Aircraft | Caltech101 | Cars | DTD | EuroSAT | Flower102 | Food101 | Pets | SUN397 | UCF101 | *Mean* |
|---|---|---|---|---|---|---|---|---|---|---|---|
| CLIP-ResNet-50 | 16.11 | 87.26 | 55.89 | 40.37 | 25.79 | 62.77 | 74.82 | 82.97 | 60.85 | 59.48 | 56.63 |
| TPT (Shu et al., 2022) | 17.58 | 87.02 | 58.46 | 40.84 | 28.33 | 62.69 | 74.88 | 84.49 | 61.46 | 60.82 | 57.66 |
| C-TPT (Yoon et al., 2024) | 17.50 | 87.40 | 57.30 | 43.10 | 29.40 | 65.30 | 76.00 | 84.00 | 62.10 | 60.70 | 58.28 |
| HisTPT (Tang et al., 2023) | 18.10 | 87.20 | **61.30** | 41.30 | 42.50 | 67.60 | **81.30** | 84.90 | 63.50 | 64.10 | 61.18 |
| TDA (Karmanov et al., 2024) | 17.61 | 89.70 | 57.78 | 43.74 | 42.11 | **68.74** | 77.75 | 86.18 | 62.53 | 64.18 | 61.03 |
| BCA (Zhou et al., 2025) | **19.89** | 89.70 | 58.13 | **48.58** | 42.12 | 66.30 | 77.19 | 85.58 | 63.38 | 63.51 | 61.44 |
| *Rose-TTA (Ours)* | 18.46 | 89.73 | 58.05 | 45.10 | 49.31 | 68.17 | 77.78 | 86.51 | 63.41 | 64.23 | **62.07** |
| CLIP-ViT-B/16 | 23.22 | 93.55 | 66.11 | 45.04 | 50.42 | 66.99 | 82.86 | 86.92 | 65.63 | 65.16 | 64.59 |
| TPT (Shu et al., 2022) | 24.78 | 94.16 | 66.87 | 47.75 | 42.44 | 68.98 | 84.67 | 87.79 | 65.50 | 68.04 | 65.10 |
| C-TPT (Yoon et al., 2024) | 23.90 | 94.10 | 66.70 | 46.80 | 48.70 | 69.90 | 84.50 | 87.40 | 66.00 | 66.70 | 65.47 |
| MTA (Zanella & Ben Ayed, 2024) | 25.20 | 94.21 | 68.47 | 45.90 | 45.36 | 68.06 | 85.00 | 88.24 | 66.67 | 68.69 | 65.58 |
| PromptAlign (Karmanov et al., 2023) | 24.80 | 94.01 | **68.50** | 47.24 | 47.86 | 72.39 | 86.65 | 90.76 | 67.54 | 69.47 | 66.92 |
| HisTPT (Tang et al., 2023) | 26.90 | 94.50 | 69.20 | 48.90 | 49.70 | 71.20 | **89.30** | 89.10 | 67.20 | 70.10 | 67.61 |
| TDA (Karmanov et al., 2024) | 23.91 | 94.24 | 67.28 | 47.40 | 58.00 | 71.42 | 86.14 | 88.63 | 67.62 | 70.66 | 67.53 |
| BCA (Zhou et al., 2025) | **28.59** | 94.69 | 66.86 | **53.49** | 56.63 | 73.12 | 85.97 | 90.43 | **68.41** | 67.59 | 68.58 |
| *Rose-TTA (Ours)* | 25.86 | **94.76** | 67.06 | 48.00 | **65.64** | **74.17** | 86.20 | **90.95** | 68.00 | **71.34** | **69.20** |

Table 2: **Comparisons on the out-of-distribution benchmark.** ROSE-TTA achieves best overall performance and surpasses other alternatives on three of the five datasets. The best and runner-up results are bolded and underlined, respectively.

| Method | ImageNet | ImageNet-V2 | ImageNet-S | ImageNet-A | ImageNet-R | *Mean* |
|---|---|---|---|---|---|---|
| CLIP-ResNet-50 (Radford et al., 2021) | 59.81 | 52.91 | 35.48 | 23.24 | 60.72 | 46.43 |
| TPT (Shu et al., 2022) | 60.74 | 54.70 | 35.09 | 26.67 | 59.11 | 47.62 |
| C-TPT (Yoon et al., 2024) | 61.20 | 54.80 | 35.70 | 25.60 | 59.70 | 47.40 |
| TDA (Karmanov et al., 2024) | 61.35 | 55.54 | **38.12** | 30.29 | 62.58 | 49.57 |
| BCA (Zhou et al., 2025) | 61.81 | **56.58** | 38.08 | 30.35 | 62.89 | 49.94 |
| *Rose-TTA(Ours)* | **62.00** | 56.45 | 37.96 | **30.93** | **63.01** | **50.07** |
| CLIP-ViT-B/16 | 68.34 | 61.88 | 48.24 | 49.89 | 77.65 | 61.20 |
| TPT (Shu et al., 2022) | 68.98 | 63.45 | 47.94 | 54.77 | 77.06 | 62.44 |
| C-TPT (Yoon et al., 2024) | 69.30 | 63.40 | 48.50 | 52.90 | 78.00 | 62.42 |
| MTA (Zanella & Ben Ayed, 2024) | 70.08 | 64.24 | 49.61 | 58.06 | 78.33 | 64.06 |
| PromptAlign (Karmanov et al., 2023) | - | 65.29 | 50.23 | 59.37 | 79.33 | 63.56 |
| TDA (Karmanov et al., 2024) | 69.51 | 64.67 | 50.54 | 60.11 | 80.24 | 65.01 |
| BCA (Zhou et al., 2025) | 70.22 | 64.90 | **50.87** | 61.14 | 80.72 | 65.57 |
| *Rose-TTA(Ours)* | **70.54** | **65.83** | 50.82 | **61.22** | **81.81** | **66.04** |

overall performance across the ten datasets compared with other state-of-the-art methods. Compared with gradient-based prompt tuning methods (Shu et al., 2022; Tang et al., 2023; Yoon et al., 2024), our method performs better on eight of the ten datasets. Moreover, since ROSE-TTA avoids gradient computation and backpropagation, it is also more efficient than the prompt tuning methods. Our method also outperforms recent cache-based efficient adaption methods (Karmanov et al., 2024; Zhou et al., 2025) on six of the ten datasets. On ViT-B/16, our method again delivers the best overall results, leading on five of the ten datasets. The results demonstrate that our method effectively adapts across diverse datasets.

**Results on the OOD benchmark.** We also evaluate ROSE-TTA on the OOD benchmark (Table 2), with conclusions consistent with the cross-dataset results. Our method surpasses both prompt tuning approaches and recent cache-based approaches in the overall accuracy, achieving the best performance on ImageNet, ImageNet-A, ImageNet-V2, and ImageNet-R. The findings demonstrate the effectiveness of ROSE-TTA for adaptation across different data distributions.

## 5.2 ABLATION STUDY

**Component analysis.** To validate the effectiveness of each proposed component, we perform ablation studies by systematically removing individual modules from ROSE-TTA. The experiments are conducted on both OOD and cross-dataset benchmarks based on ViT/B-16. As shown

Table 3: **Ablations on components for cache predictions.** Removing $\mathbf{d}$ in Eq. (4), $P_t$ and $P_i$ in Eq. (10), or $\hat{F}$ in Eq. (8) leads to performance degradation.

| Dataset | w/o $\mathbf{d}$ | w/o $P_t, P_i$ | w/o $\hat{F}$ | **Rose-TTA** |
|---|---|---|---|---|
| Cross-Dataset | 67.89 | 68.24 | 68.78 | **69.20** |
| OOD | 65.03 | 65.22 | 65.68 | **66.04** |

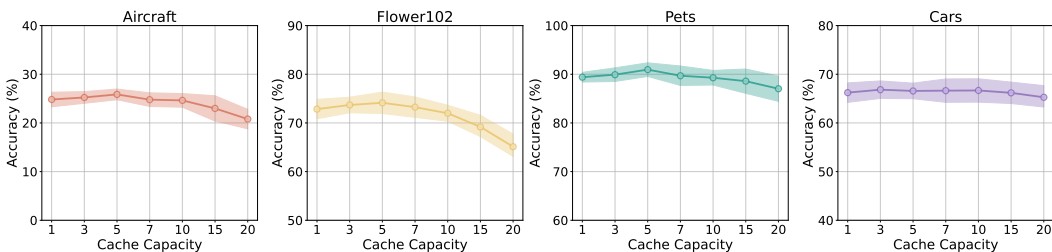

Figure 3: **Analysis on cache capacity** $n_{\mathcal{C}}$**.** Our method performs best with $n_{\mathcal{C}}$ as 5 for most datasets.

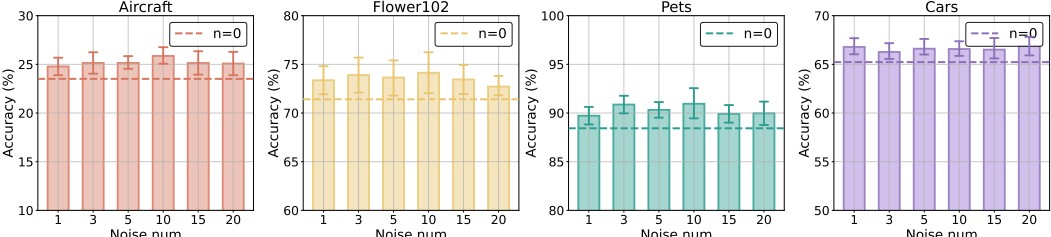

Figure 4: **Analysis on the number of noise-augmented features** $n$**. Our performance is relatively stable across** $n$**, with the best results around 10 and better than** $n = 0$**.**

in Table 3, without the noise-based prediction stability $\mathbf{d}(\boldsymbol{x})$ for cache construction, the cache can only select samples according to the entropy value, which can be unreliable and lead to performance degradation. When removing $P_t$ and $P_i$ from the final prediction, there is no class information completion for the cache utilization, resulting performance degradation. Similarly, without the sample-specific refined cache features $\hat{\boldsymbol{F}}$, relying only on static cache prevents adaptation to individual test samples, highlighting the importance of our dynamic pseudo-gradient updates.

**Inference time comparison.** In this experiment, we evaluated the efficiency and effectiveness of the proposed Rose-TTA method on the ImageNet dataset using Vit-16 as the visual backbone on RTX 4090 GPU. We compared ours with TPT, DiffTPT and TDA, the results are shown in Table 4. Our method delivers 26× faster inference than DiffTPT with +0.24% accuracy, and 12× faster than TPT with +1.56% gain. The

Table 4: **Efficiency and accuracy comparison on ImageNet.** Our method delivers obvious speedup while maintaining high accuracy.

| Method | Time (min) | Memory (MB) | Accuracy (%) | Gain (%) |
|---|---|---|---|---|
| CLIP | 9.36 | 808.96 | 68.34 | — |
| TPT | 585.00 | 4392.96 | 68.98 | 0.64 |
| DiffTPT | 1272.00 | 4710.40 | 70.30 | 1.96 |
| TDA | 50.00 | 860.16 | 69.51 | 1.17 |
| ROSE-TTA | 48.96 | 1234.32 | 70.54 | 2.20 |

speedup comes from our training-free design that avoids backward passes and prompt optimization. Unlike TDA, which uses two caches, ROSE-TTA uses a single noise-aware cache, achieving +1.03% performance and 1.02× speedup. Further details on efficiency are provided in Appendix B.1.

**Effect of Cache Capacity** $n_{\mathcal{C}}$**.** To assess the impact of cache capacity, i.e., the number of historical key-value pairs per class, we conduct experiments by varying it from 1 to 20 on four datasets based on ViT/B-16. As shown in Figure 3, ROSE-TTA performs best usually with a cache size 5, while it maintaining relatively stable accuracy on most datasets. This stability suggests that our structural completion and replacement strategies effectively regulate cache content. However, some datasets like Flower102 exhibit more pronounced fluctuations, especially with larger capacities. The reason can be its fine-grained class structure, where categories share highly similar visual patterns. In such cases, an overly large cache risks accumulating redundant or noisy samples that blur decision boundaries. Moreover, under online adaptation, large capacities can also amplify the effect of early, less reliable entries, introducing additional instability.

**Impact of Noise Augmentation Number** $n$**.** We also ablate the number of noise augmentations used for stability estimation in cache construction. The experiments are also conducted on four datasets based on ViT/B-16. As shown in Figure 4, performance rises slightly with more augmentations, with a peak around 10. Compared with $n = 0$, noise augmentation obviously improves classification performance in most cases. This indicates that the uncertainty-aware mechanism in ROSE-TTA is robust, yeilding reliable stability estimation with even small numbers of augmentations (e.g., 3).

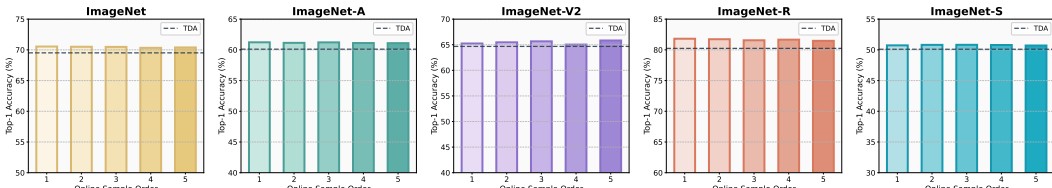

Figure 6: **Sensitivity to test-time sample order on five OOD datasets.** The performance are stable across datasets, demonstrating the robustness of the algorithm to different test example orders.

While increasing augmentations slightly improves early stability, excessive augmentation tends to yield diminishing returns and may introduce redundant signals, thereby hindering the model's ability to capture genuine data characteristics.

**Progressive adaptation analysis.** To further understand the dynamic behavior of ROSE-TTA during online adaptation, we evaluate the method at multiple test-time checkpoints on ImageNet and EuroSAT. The results are provided in Figure 5. Compared with TDA and BCA, our method consistently achieves higher accuracy throughout the entire process. Notably, advantage is more pronounced at the early stage (e.g., at 10%–30% on-

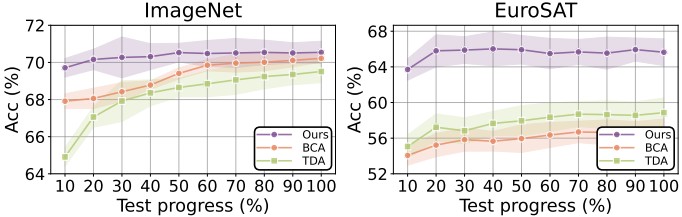

Figure 5: **Progressive adaptation analysis** on ImageNet and EuroSAT. Our method consistently outperforms TDA and BCA while more pronounced at the early stages, indicating that the method caches higher-quality samples and mitigates class imbalance by information completion.

line data), where TDA suffers from unstable performance while ROSE-TTA maintains good improvement. Despite not depending on entropy for prediction, BCA cannot avoid the early-stage performance collapse in the absence of a proper selection mechanism for high-quality samples. The results indicate that our information completion and uncertainty-aware cache construction provide reliable guidance from the beginning of adaptation, caching higher-quality samples and alleviating class imbalance. As adaptation progresses, the performance gap narrows but remains, suggesting that ROSE-TTA not only ensures robustness in the initial phase, but also preserves long-term effectiveness throughout the evaluation.

**Robustness to random sample ordering.** As our method updates the cache online, the performance can be influenced by the order of the test samples. To show the robustness of our method regarding the test sample orders, we conducted multiple experiments on the OOD dataset with different sample orders for 5 rounds. We observe that there are fluctuations in the performance of both datasets. The experimental results in Figure 6 demonstrate that our method performs stably under random order conditions, confirming the robustness of the algorithm to the random permutation of test examples. Moreover, independent of the test order, the proposed method surpasses TDA consistently.

## 6 CONCLUSION

In this work, we propose ROSE-TTA, a reliable, test-time–enhanced caching framework that improves CLIP's zero-shot performance across datasets and distribution shifts. ROSE-TTA construct a more reliable cache by selecting more stable test features via a noise-aware uncertainty strategy that integrates entropy minimization with noise-based stability. During cache utilization, we enhance its global class information through a graph-based structural completion strategy that mitigates class imbalance and strengthens cache representations. The method also injects local instance information with a sample-specific refinement module to adapt cached features to each incoming test sample. Extensive experiments on 15 datasets demonstrate the effectiveness of ROSE-TTA, providing a robust and efficient approach to test-time cache construction and utilization in practice.

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

## USE OF LARGE LANGUAGE MODELS (LLMS)

We use LLMs (e.g., ChatGPT) only for minor language polishing. They did not contribute to research ideation, experimental design, or substantive writing.

## ETHICS STATEMENT

Our contribution is primarily about the online cache for test-time adaptation in vision-language models. The method does not introduce ethical risks.

## REPRODUCIBILITY STATEMENT

We provide the sufficient datasets and implementation details in Section 5 for reproducibility.

## APPENDIX

## A   COMPARISON OF THE PSEUDO-GRADIENT WITH ACTUAL GRADIENT

### A.1   THEORETICAL ANALYSIS: PSEUDO-GRADIENT APPROXIMATION

**Connection to entropy minimization.** The common gradient-based method for test-time adaptation calculates gradients based on entropy minimization $H(p) = -\sum_{k=1}^{C} p_k \log p_k$, where $p_k$ denotes the predicted probability for class $k$ and $C$ is the number of classes. In our sample-specific cache refinement (Eq. 8), the pseudo gradient consists of two components: $\gamma = 1 - \mathcal{H}(P_n)$ and $\zeta = P_n - \frac{1}{C}$, where $P_n = \text{softmax}\left(\frac{1}{n}\sum_{i=1}^{n} P_{cache}(\tilde{\boldsymbol{f}}_{\boldsymbol{x}}^i)\right)$ is the averaged cache prediction based on noise-augmented features and $\mathcal{H}(\cdot)$ denotes the normalized entropy. Here, $\gamma$ controls the update intensity (higher confidence yields stronger updates), and $\zeta$ controls the update direction. This design is motivated by the behavior of entropy minimization, which pushes probabilities above the uniform distribution higher and pulls those below uniform lower.

In cache-based TTA, the prediction for class $c$ is computed via the similarity between the test feature $\boldsymbol{f}_{\boldsymbol{x}}$ and cached features $\{f_i\}_{i=1}^{N}$ in $\boldsymbol{F}$. For a cached feature $f_c$ belonging to class $c$, the corresponding logit is $z_c = \boldsymbol{f}_{\boldsymbol{x}}^T f_c$, and $p_c$ is the probability after softmax. The gradient of entropy with respect to $f_c$ can be decomposed as:

$$\frac{\partial H}{\partial f_c} = \frac{\partial H}{\partial p_c}\frac{\partial p_c}{\partial z_c}\frac{\partial z_c}{\partial f_c} = \frac{\partial H}{\partial p_c}\frac{\partial p_c}{\partial z_c}\boldsymbol{f}_{\boldsymbol{x}}, \tag{11}$$

where $\frac{\partial z_c}{\partial f_c} = \boldsymbol{f}_{\boldsymbol{x}}$ follows from the inner product $z_c = \boldsymbol{f}_{\boldsymbol{x}}^T f_c$.

The first term is $\frac{\partial H}{\partial p_c} = \frac{\partial}{\partial p_c}\left(-\sum_{k=1}^{C} p_k \log p_k\right) = -\log p_c - 1$, serving as the directional component that determines how each probability should be adjusted. The second term stems from the softmax Jacobian: for softmax $p_c = \frac{e^{z_c}}{\sum_k e^{z_k}}$, we have

$$\frac{\partial p_c}{\partial z_c} = p_c(1 - p_c). \tag{12}$$

Critically, since $p_c \in (0, 1)$, this term is always positive and only modulates the update magnitude without affecting its direction. Therefore, the directional behavior of entropy minimization is determined by $\frac{\partial H}{\partial p_c} = -\log p_c - 1$, while the update naturally includes the test feature $\boldsymbol{f_x}$ as a multiplicative factor.

**Linear approximation via Taylor expansion.** The nonlinearity in $-\log p_c - 1$ can be numerically unstable when probabilities approach 0 or 1, especially under distribution shifts. We derive a stable linear approximation by performing first-order Taylor expansion around the uniform distribution $p_c = \frac{1}{C}$:

$$-\log p_c - 1 \approx -\log \frac{1}{C} - 1 + \frac{d}{dp_c}\left(-\log p_c - 1\right)\bigg|_{p_c = \frac{1}{C}} \left(p_c - \frac{1}{C}\right). \qquad (13)$$

Computing the derivative: $\frac{d}{dp_c}(-\log p_c - 1) = -\frac{1}{p_c}$, which evaluated at $p_c = \frac{1}{C}$ gives $-C$. The constant term $-\log \frac{1}{C} - 1 = \log C - 1$ does not affect the gradient direction and can be absorbed into the pseudo learning rate $\eta$. Thus:

$$-\log p_c - 1 \approx -C\left(p_c - \frac{1}{C}\right) + \text{const}. \qquad (14)$$

Combining with the test feature factor, the gradient direction is approximated as $-C(p_c - \frac{1}{C}) \cdot \boldsymbol{f_x}$. Therefore, **our cache update in Eq. (8): $\hat{F} = F + \eta \cdot \gamma \cdot \zeta \cdot \boldsymbol{f_x}$ directly implements this approximation**.

Table 5: Comparison of pseudo-gradient vs true gradient

| Method | Aircraft | Caltech101 | Cars | DTD | EuroSAT | Flower102 | Food101 | Pets | SUN397 | UCF101 | ImageNet | ImageNet-A | ImageNet-V2 | ImageNet-R | ImageNet-S | Mean |
|---|---|---|---|---|---|---|---|---|---|---|---|---|---|---|---|---|
| Pseudo-gradient | 25.86 | 94.76 | 67.06 | 48.00 | 65.64 | 74.17 | 86.20 | 90.95 | 68.00 | 71.34 | 70.54 | 61.22 | 65.83 | 81.81 | 50.82 | 68.15 |
| True gradient | 26.73 | 94.93 | 66.92 | 48.78 | 66.01 | 74.25 | 88.13 | 90.46 | 67.43 | 71.23 | 71.11 | 61.11 | 65.9 | 81.33 | 51.32 | 68.38 |

## A.2 EMPIRICAL RESULTS

We also conducted experiments using true-gradient refinement based on entropy minimization. The results in Table 5 are provided in the following table: Across all datasets tested, we found that the performance of the pseudo-gradient closely matches that of the true entropy-minimization gradient with less computational cost, empirically demonstrating the proposed pseudo-gradient.

## B MORE EFFICIENCY DETAILS

### B.1 COMPUTATIONAL EFFICIENCY ANALYSIS OF DIFFERENT COMPONENT

We profile the runtime and memory footprint of each module during inference, as shown in Table 6.

Table 6: Breakdown of computational costs for each component during test-time adaptation.

| Component | Time (min) | Memory (MB) |
|---|---|---|
| Noise Augmentation | 1.57 | 209.27 |
| Graph Construction | 5.82 | 310.01 |
| Sample-Specific Adapt. | 1.41 | 22.89 |
| **Overall** | **48.96** | **1234.33** |

As Table 6 reveals, our efficiency stems from several principled design choices that collectively minimize computational overhead. Unlike TPT (Shu et al., 2022) and DiffTPT (Feng et al., 2023),

which require iterative gradient descent over prompts at test time, our method operates entirely in the forward pass by leveraging pre-computed prototypes and graph-based reasoning, thereby eliminating the computational burden of backward propagation with the use of pesudo gradient The noise augmentation module, while generating diverse views of each sample, incurs only a one-time cost of 0.57 minutes as these augmentations are fully reusable across the test set, reducing per-sample overhead to negligible levels (<0.01 min/sample) and accounting for merely 1.2% of total runtime. Most notably, despite involving multiple matrix operations for edge construction and message passing, our graph module adds only 11.9% overhead (5.82 min) through highly parallelizable operations that exploit modern GPU architectures, enabling rich structural reasoning without sacrificing speed.

## B.2 COMPUTATIONAL EFFICIENCY WITH VARYING NOISE BUDGETS

Table 7 demonstrates the scalability of our noise-enhanced uncertainty estimation on ImageNet. Increasing the noise budget from $n = 1$ to $n = 20$ introduces only 1.11 minutes (2.28%) additional runtime. This remarkably low overhead validates our design: the computational cost is dominated by one-time noise pre-generation rather than per-sample operations. Our default choice of $n = 5$ balances stability estimation quality with minimal overhead (48.90 min, 1131 MB), while more conservative settings ($n = 3$) or aggressive ones ($n = 10$) remain viable depending on application requirements.

Table 7: Scalability with varying noise budgets

| Noise Num | Testing Time (ms/sample) | Memory Usage (KB/sample) |
|:---:|:---:|:---:|
| 1 | 58.36 | 21.40 |
| 3 | 58.51 | 22.29 |
| 5 | 58.68 | 23.17 |
| 10 | 58.75 | 25.28 |
| 15 | 59.17 | 27.53 |
| 20 | 59.69 | 29.64 |

## C    EXTRA EXPERIMENT RESULTS

### C.1    MORE EXPERIMENTS ON STANDARD OOD CORRUPTION BENCHMARKS

To show the effectiveness of the proposed method on more benchmarks, we have incorporated additional experiments on CIFAR-10-C, CIFAR-100-C, and ImageNet-C. We reproduced zero-shot CLIP and TDA on the datasets for comparisons. The results are in the following Table 8. Our **ROSE-TTA** achieves better overall performance on these three datasets compared with CLIP, TDA and BCA, demonstrating the effectiveness of our method on these challenging distribution shifts.

### C.2    COMPARISONS OF CACHE ACCURACY

As shown in Figure 7, on both FGVC and ImageNet-A datasets, our proposed ROSE-TTA consistently achieves higher cache accuracy than the baseline TDA method across both early and final stages of testing. Specifically, the distribution of cache accuracy is noticeably shifted toward higher values under ROSE-TTA, indicating that our method effectively enhances the reliability of cached samples. This improvement not only mitigates early-stage performance collapse but also maintains strong performance throughout the adaptation process, demonstrating the robustness and practical efficacy of ROSE-TTA in scenarios with class imbalance.

### C.3    ABLATION STUDY OF $\alpha$

To further investigate the impact of the hyperparameter $\alpha$, we have conducted additional experiments across all datasets to evaluate the performance under different values of $\alpha$ varying in the range of {0,

Table 8: Mean accuracy (%) on CIFAR-10C, CIFAR-100C, and ImageNet-C - TTA mean accuracy of the 15 corruptions at severity level 5.

| Method | Gaussian | Shot | Impulse | Defocus | Glass | Motion | Zoom | Snow | Frost | Fog | Brightness | Contrast | Elastic | Pixelate | JPEG | Mean |
|---|---|---|---|---|---|---|---|---|---|---|---|---|---|---|---|---|
| **ImageNet-C** | | | | | | | | | | | | | | | | |
| CLIP | 18.56 | 12.11 | 19.45 | 31.22 | 23.12 | 32.15 | 26.25 | 34.59 | 34.17 | 45.35 | 57.11 | 30.94 | 48.93 | 45.01 | 35.14 | 32.94 |
| TDA | 19.54 | 13.46 | 19.90 | 32.76 | 26.35 | 34.32 | 27.79 | 36.42 | 35.47 | 47.79 | 58.16 | 31.92 | 49.22 | 45.05 | 35.80 | 34.26 |
| BCA | 19.76 | 13.44 | 20.38 | 33.08 | 27.84 | 34.62 | 27.88 | 38.12 | 35.96 | 52.26 | 59.02 | 32.38 | 49.20 | 45.08 | 35.94 | 34.99 |
| Ours | **20.18** | **13.57** | **21.56** | **33.67** | **29.78** | **35.19** | **28.12** | **40.17** | **36.66** | **60.89** | **59.94** | **33.26** | **49.38** | **45.18** | **36.21** | **36.25** |
| **CIFAR-10C** | | | | | | | | | | | | | | | | |
| CLIP | 46.22 | 50.17 | 45.23 | 65.45 | 43.33 | 60.14 | 68.54 | 69.12 | 71.11 | 55.23 | 80.45 | 44.53 | 55.25 | 52.17 | 60.31 | 57.82 |
| TDA | 48.66 | 53.24 | 46.26 | 67.09 | 43.55 | 63.17 | 69.97 | 70.02 | 72.57 | 56.13 | 81.90 | 45.59 | 56.84 | 53.71 | 60.91 | 59.31 |
| BCA | 49.08 | 53.72 | 46.18 | 68.00 | 44.55 | 65.92 | 70.82 | 71.38 | 72.98 | 56.75 | 81.98 | 47.05 | 56.35 | 54.02 | 61.52 | 60.03 |
| Ours | **50.14** | **54.76** | **46.31** | **69.73** | **46.17** | **69.56** | **72.45** | **73.52** | **74.23** | **57.86** | 81.96 | **49.23** | **56.97** | **54.75** | **62.69** | **61.36** |
| **CIFAR-100C** | | | | | | | | | | | | | | | | |
| CLIP | 28.15 | 28.18 | 20.17 | 38.22 | 20.85 | 35.24 | 42.56 | 40.14 | 42.88 | 30.13 | 52.19 | 20.89 | 29.66 | 24.15 | 33.11 | 32.43 |
| TDA | 29.18 | 29.84 | 24.68 | 39.44 | 20.99 | 37.44 | 43.84 | 42.86 | 44.24 | 30.72 | 53.72 | 22.85 | 30.78 | 25.13 | 33.81 | 33.97 |
| BCA | 29.66 | 30.10 | 25.96 | 40.36 | 21.64 | 37.70 | 43.62 | 42.92 | 46.28 | 30.79 | 54.90 | 23.74 | 32.16 | 25.18 | 33.88 | 34.51 |
| Ours | **30.22** | **30.46** | **27.89** | **41.58** | **22.45** | **38.11** | **44.19** | **43.00** | **48.69** | **30.87** | **56.24** | **24.78** | **33.94** | **25.22** | **33.96** | **35.44** |

1, 3, 5, 10} . Our results in Table 9 show that the performance remains relatively stable with changes in $\alpha$ and significantly outperforms the case where $\alpha$ is set to 0. Notably, across the majority of $\alpha$ values tested, our method consistently outperforms BCA, demonstrating that even with randomly selected $\alpha$, our approach maintains its effectiveness. This robustness validates the effectiveness of our cache-based refinement strategy and reduces the need for extensive hyperparameter tuning.This confirms that our cache-based refined cache prediction approach is effective.

Table 9: Performance under different $\alpha$ values

| Dataset | Aircraft | Caltech101 | Cars | DTD | EuroSAT | Flower102 | Food101 | Pets | SUN397 | UCF101 | ImageNet | ImageNet-A | ImageNet-V2 | ImageNet-R | ImageNet-S | Mean |
|---|---|---|---|---|---|---|---|---|---|---|---|---|---|---|---|---|
| $\alpha = 0$ | 24.70 | 94.16 | 65.80 | 44.50 | 47.67 | 71.42 | 86.10 | 89.15 | 66.62 | 66.77 | 70.04 | 59.43 | 64.35 | 80.33 | 49.40 | 65.36 |
| BCA | **28.59** | 94.69 | 66.86 | 53.59 | 56.63 | 73.12 | 85.97 | 90.43 | **68.41** | 67.59 | 70.22 | 61.14 | 64.90 | 80.72 | 50.87 | 67.58 |
| $\alpha = 1$ | 24.77 | 94.45 | 66.92 | 47.78 | 60.12 | 73.42 | 86.12 | **90.95** | 67.43 | **71.34** | 70.11 | 60.11 | 64.90 | 81.33 | 50.32 | 67.40 |
| $\alpha = 3$ | 25.86 | 94.50 | **67.06** | **48.00** | **65.64** | 73.67 | 86.15 | 90.42 | **68.00** | 70.33 | **70.54** | **61.22** | 65.68 | 81.56 | 50.45 | **67.94** |
| $\alpha = 5$ | 25.23 | **94.76** | 66.77 | 47.90 | 65.50 | **74.17** | 86.16 | 90.56 | 67.56 | 69.57 | 70.33 | 60.58 | 65.77 | **81.681** | **50.82** | 67.83 |
| $\alpha = 10$ | 25.69 | 94.68 | 67.02 | 47.22 | 63.25 | 74.11 | **86.16** | 90.66 | 67.78 | 70.12 | 70.42 | 61.01 | 65.83 | 81.23 | 50.64 | 67.73 |

## C.4 EMPIRICAL RESULTS OF $\sigma$

As shown in Table 10, we conducted a sensitivity analysis on $\sigma \in \{0, 0.05, 0.1, 0.15, 0.2\}$ across all datasets, where $\sigma = 0$ corresponds to using entropy only. The performance is relatively the best around $\sigma = 0.1$. If $\sigma$ is too small, the perturbations become negligible, failing to meaningfully test the sample's robustness. If $\sigma$ is too large, excessive noise can corrupt the semantic information in the features, causing even truly reliable samples to exhibit unstable predictions. This validates our choice of $\sigma = 0.1$ as a robust default.

## C.5 PERFORMANCE GAINS COMPARISON WITH VARYING NOISE LEVEL

In Table 11, we have added the comparisons with the no-noise-augmentation baseline ($n = 0$) to more clearly demonstrate the benefits of noise augmentation. Below, we present the performance of each dataset at different noise levels, where "Noise Gains" represents the average performance gain relative to the no-noise baseline. As shown, noise augmentation obviously improves classification performance in most cases, and our method exhibits good stability across different noise levels.

Table 10: Performance under different $\sigma$ values

| $\sigma$ | Cross Dataset Avg | OOD Avg |
|---|---|---|
| 0 | 68.21 | 65.13 |
| 0.05 | 69.17 | 66.01 |
| 0.1 | **69.20** | **66.04** |
| 0.15 | 69.11 | 65.98 |
| 0.2 | 68.88 | 65.56 |

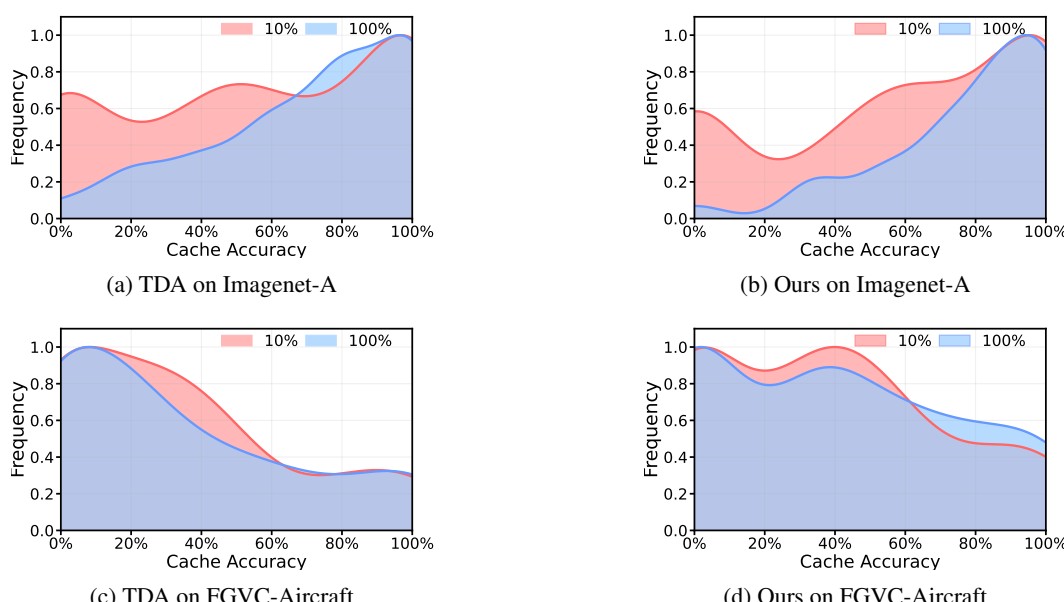

(a) TDA on Imagenet-A

(b) Ours on Imagenet-A

(c) TDA on FGVC-Aircraft

(d) Ours on FGVC-Aircraft

Figure 7: Cache accuracy comparisons across datasets and methods.

## C.6 CACHE DYNAMICS ANALYSIS

Table 14 reveals the effectiveness of our doubly robust cache construction. Across all 15 datasets, the average replacement ratio is only **12.7%**, indicating that cached samples remain highly stable throughout the online testing process. This low replacement rate validates that our noise-enhanced uncertainty estimation effectively identifies truly representative prototypes that persist under distribution shifts. Notably, the replacement behavior exhibits dataset-dependent adaptivity. ImageNet variants (A/V2/R/S) show moderate replacement ratios of 11-18%, reflecting our cache's ability to adapt appropriately to different types and severities of natural corruptions while maintaining core prototypical features.

## C.7 CACHE REPLACEMENT POLICY ANALYSIS

To validate our eviction design, we conducted ablation studies comparing ROSE-TTA's noise-aware joint scoring against common cache replacement policies: (1) Entropy-only, (2) Similarity, (3) Random, (4) FIFO (First-In-First-Out), (5) LRU (Least Recently Used). The results in Table 12 reveal critical insights. The poor performance of FIFO (64.68%/62.25%) and LRU (63.21%/60.02%) validates that early-arriving samples are not necessarily more representative, especially when test data arrives in random order.

Entropy-only (68.21%/65.13%) performs competitively but still accumulates misclassified confident samples. Our dual-criterion approach achieves consistent improvements across both settings, with notably larger gains on OOD data (+0.91%). This validates our hypothesis: under severe distribution shifts, combining entropy-based confidence with noise-enhanced stability provides complementary signals that effectively filter unreliable samples while retaining genuinely robust prototypes. Moreover, we provide a comprehensive analysis of replacement frequency and its associated computational

Table 11: Impact of noise augmentation budget on accuracy. We report accuracy ± std and gain ± std over $n = 0$ baseline.

| Dataset | $n = 0$ Acc | $n = 1$ Acc | Gain | $n = 3$ Acc | Gain | $n = 5$ Acc | Gain | $n = 10$ Acc | Gain | $n = 15$ Acc | Gain | $n = 20$ Acc | Gain |
|---|---|---|---|---|---|---|---|---|---|---|---|---|---|
| Aircraft | $23.51_{\pm.41}$ | $24.78_{\pm.82}$ | $1.27_{\pm.23}$ | $25.14_{\pm.77}$ | $1.63_{\pm.24}$ | $25.16_{\pm.12}$ | $1.65_{\pm.11}$ | $25.86_{\pm.14}$ | $\mathbf{2.35}_{\pm.11}$ | $25.14_{\pm.79}$ | $1.63_{\pm.22}$ | $25.11_{\pm.78}$ | $1.60_{\pm.25}$ |
| Caltech101 | $94.16_{\pm.01}$ | $94.52_{\pm.03}$ | $0.36_{\pm.01}$ | $94.62_{\pm.05}$ | $0.46_{\pm.02}$ | $94.63_{\pm.01}$ | $0.47_{\pm.01}$ | $94.76_{\pm.05}$ | $\mathbf{0.60}_{\pm.03}$ | $94.61_{\pm.02}$ | $0.45_{\pm.01}$ | $94.59_{\pm.02}$ | $0.43_{\pm.01}$ |
| Cars | $65.73_{\pm.23}$ | $66.78_{\pm.76}$ | $1.05_{\pm.33}$ | $66.27_{\pm.74}$ | $0.54_{\pm.43}$ | $66.61_{\pm.66}$ | $0.88_{\pm.56}$ | $67.06_{\pm.54}$ | $\mathbf{1.33}_{\pm.64}$ | $66.57_{\pm.45}$ | $0.84_{\pm.35}$ | $66.91_{\pm.44}$ | $1.18_{\pm.52}$ |
| DTD | $44.61_{\pm.39}$ | $47.34_{\pm.76}$ | $2.73_{\pm.63}$ | $47.38_{\pm.89}$ | $2.77_{\pm.77}$ | $47.56_{\pm.98}$ | $2.95_{\pm.82}$ | $48.00_{\pm.78}$ | $\mathbf{3.39}_{\pm.67}$ | $47.64_{\pm.33}$ | $3.03_{\pm.23}$ | $47.79_{\pm.56}$ | $3.18_{\pm.55}$ |
| EuroSAT | $58.67_{\pm.24}$ | $64.25_{\pm1.01}$ | $5.58_{\pm.78}$ | $64.86_{\pm.74}$ | $6.19_{\pm.69}$ | $64.34_{\pm.56}$ | $5.67_{\pm.73}$ | $65.64_{\pm.62}$ | $\mathbf{6.97}_{\pm.54}$ | $65.12_{\pm.29}$ | $6.45_{\pm.13}$ | $65.32_{\pm.27}$ | $6.65_{\pm.09}$ |
| Flower102 | $71.41_{\pm.23}$ | $73.37_{\pm1.45}$ | $1.96_{\pm.79}$ | $73.89_{\pm1.32}$ | $2.48_{\pm1.23}$ | $73.65_{\pm1.57}$ | $2.24_{\pm1.21}$ | $74.17_{\pm1.02}$ | $\mathbf{2.76}_{\pm.98}$ | $73.44_{\pm1.23}$ | $2.03_{\pm1.01}$ | $72.71_{\pm.99}$ | $1.30_{\pm.45}$ |
| Food101 | $86.11_{\pm.01}$ | $86.18_{\pm.00}$ | $0.07_{\pm.01}$ | $86.17_{\pm.01}$ | $0.06_{\pm.01}$ | $86.15_{\pm.03}$ | $0.04_{\pm.02}$ | $86.21_{\pm.00}$ | $\mathbf{0.10}_{\pm.01}$ | $86.20_{\pm.01}$ | $0.09_{\pm.01}$ | $86.19_{\pm.03}$ | $0.08_{\pm.02}$ |
| Pets | $89.17_{\pm.43}$ | $89.72_{\pm.89}$ | $0.55_{\pm.42}$ | $90.87_{\pm.67}$ | $1.70_{\pm.78}$ | $90.32_{\pm.79}$ | $1.15_{\pm.72}$ | $90.95_{\pm1.01}$ | $\mathbf{1.78}_{\pm.92}$ | $89.92_{\pm.65}$ | $0.75_{\pm.52}$ | $89.97_{\pm.78}$ | $0.80_{\pm.62}$ |
| SUN397 | $66.62_{\pm.12}$ | $67.88_{\pm.13}$ | $1.26_{\pm.08}$ | $67.76_{\pm.24}$ | $1.14_{\pm.15}$ | $67.54_{\pm.03}$ | $0.92_{\pm.07}$ | $68.00_{\pm.15}$ | $\mathbf{1.38}_{\pm.14}$ | $67.36_{\pm.11}$ | $0.74_{\pm.08}$ | $67.68_{\pm.15}$ | $1.06_{\pm.21}$ |
| UCF101 | $68.12_{\pm.25}$ | $70.23_{\pm.67}$ | $2.11_{\pm.42}$ | $70.68_{\pm.56}$ | $2.56_{\pm.55}$ | $71.29_{\pm.78}$ | $3.17_{\pm.74}$ | $71.34_{\pm.89}$ | $\mathbf{3.22}_{\pm.92}$ | $71.11_{\pm.85}$ | $2.99_{\pm.93}$ | $70.98_{\pm1.01}$ | $2.86_{\pm.97}$ |
| ImageNet | $70.05_{\pm.01}$ | $70.24_{\pm.04}$ | $0.19_{\pm.04}$ | $70.48_{\pm.07}$ | $0.43_{\pm.06}$ | $70.29_{\pm.04}$ | $0.24_{\pm.03}$ | $70.54_{\pm.06}$ | $\mathbf{0.49}_{\pm.05}$ | $70.31_{\pm.05}$ | $0.26_{\pm.05}$ | $70.44_{\pm.04}$ | $0.39_{\pm.03}$ |
| ImageNet-A | $59.43_{\pm.06}$ | $60.87_{\pm.06}$ | $1.44_{\pm.01}$ | $61.15_{\pm.02}$ | $1.72_{\pm.03}$ | $60.96_{\pm.03}$ | $1.53_{\pm.03}$ | $61.22_{\pm.05}$ | $\mathbf{1.79}_{\pm.06}$ | $61.21_{\pm.02}$ | $1.78_{\pm.05}$ | $61.03_{\pm.04}$ | $1.60_{\pm.03}$ |
| ImageNet-V2 | $64.35_{\pm.21}$ | $65.11_{\pm.25}$ | $0.76_{\pm.11}$ | $65.25_{\pm.31}$ | $0.90_{\pm.31}$ | $65.76_{\pm.09}$ | $1.41_{\pm.15}$ | $65.83_{\pm.32}$ | $\mathbf{1.48}_{\pm.25}$ | $65.81_{\pm.13}$ | $1.46_{\pm.11}$ | $65.72_{\pm.23}$ | $1.37_{\pm.22}$ |
| ImageNet-R | $80.37_{\pm.13}$ | $81.54_{\pm.11}$ | $1.17_{\pm.04}$ | $81.64_{\pm.13}$ | $1.27_{\pm.05}$ | $81.79_{\pm.45}$ | $1.42_{\pm.25}$ | $81.81_{\pm.03}$ | $\mathbf{1.44}_{\pm.14}$ | $81.77_{\pm.14}$ | $1.40_{\pm.13}$ | $81.57_{\pm.09}$ | $1.20_{\pm.11}$ |
| ImageNet-S | $49.41_{\pm.02}$ | $50.67_{\pm.05}$ | $1.26_{\pm.02}$ | $50.74_{\pm.05}$ | $1.33_{\pm.03}$ | $50.72_{\pm.07}$ | $1.31_{\pm.04}$ | $50.82_{\pm.07}$ | $\mathbf{1.41}_{\pm.05}$ | $50.81_{\pm.04}$ | $1.40_{\pm.01}$ | $50.69_{\pm.03}$ | $1.28_{\pm.02}$ |

Table 12: Comparison of cache eviction strategies

| Method | Cross Dataset Avg | OOD Avg |
|---|---|---|
| Entropy only | 68.21 | 65.13 |
| Similarity | 68.13 | 64.99 |
| Random | 67.11 | 63.64 |
| FIFO | 64.68 | 62.25 |
| LRU | 63.21 | 60.02 |
| Ours | 69.02 | 66.04 |

overhead in Table 13. Our noise-aware selection achieves 43.9% fewer replacements than entropy-only, while simultaneously delivering faster inference and higher accuracy.

In contrast, cross-dataset benchmarks display more diverse replacement patterns: fine-grained datasets like EuroSAT (0.01) and UCF101 (0.03) exhibit minimal replacement, suggesting that their test distributions closely align with initial cache entries, whereas coarse-grained datasets like Flower102 (0.16) and Food101 (0.16) require more frequent updates to capture intra-class variability. The consistently low replacement ratios across diverse benchmarks validate that our dual-criterion selection (entropy + stability) successfully filters noisy samples while allowing necessary adaptation, leading to robust test-time performance.

## C.8 EFFECT OF TEXT EMBEDDINGS ALLEVIATE BIASED LOGITS AT EARLY STAGE

To further the effect of $W_C$ for mitigating class imbalance in the early stage, we also provide an ablation study on the predictive cache logits with and without the textual graph at 10% progress. We average all cache logits of categories with each cache capacity. As shown in the Table 16 , without the textual graph, classes with fewer samples in the cache contribute very small logits (even 0). This leads to the early inference process being heavily influenced by a few dominant classes. Additionally, classes that have not yet appeared in the cache fail to provide any useful information, exacerbating the class imbalance. By contrast, with the textual graph, our method alleviates the bias with more balancing predictive logits.

Table 13: Replacement frequency comparison

| Method | Frequency | Testing Time(min) | Memory Usage(MB) |
|---|---|---|---|
| Noise aware | 8846 | 48.96 | 1234.33 |
| Entropy only | 15762 | 49.88 | 1086.11 |

Table 14: Cache replacement statistics across different datasets

| Dataset | Method | Replacement Num | Replacement Ratio |
|---|---|---|---|
| *ImageNet Variants* | | | |
| ImageNet | TDA | 15762 | 0.31 |
| | Ours | 8846 | 0.18 |
| ImageNet-A | TDA | 1456 | 0.19 |
| | Ours | 801 | 0.11 |
| ImageNet-V2 | TDA | 2384 | 0.24 |
| | Ours | 1325 | 0.13 |
| ImageNet-R | TDA | 7825 | 0.27 |
| | Ours | 4375 | 0.15 |
| ImageNet-S | TDA | 13184 | 0.26 |
| | Ours | 7346 | 0.15 |
| *Cross-Dataset Benchmarks* | | | |
| Aircraft | TDA | 528 | 0.16 |
| | Ours | 300 | 0.09 |
| Caltech101 | TDA | 443 | 0.18 |
| | Ours | 246 | 0.10 |
| Cars | TDA | 1339 | 0.16 |
| | Ours | 744 | 0.09 |
| DTD | TDA | 258 | 0.14 |
| | Ours | 143 | 0.08 |
| EuroSAT | TDA | 118 | 0.02 |
| | Ours | 65 | 0.01 |
| Flower102 | TDA | 513 | 0.29 |
| | Ours | 285 | 0.16 |
| Food101 | TDA | 8546 | 0.28 |
| | Ours | 4753 | 0.16 |
| Pets | TDA | 314 | 0.09 |
| | Ours | 174 | 0.05 |
| SUN397 | TDA | 2178 | 0.11 |
| | Ours | 1211 | 0.06 |
| UCF101 | TDA | 236 | 0.05 |
| | Ours | 131 | 0.03 |

Table 15: Robustness to random sample ordering

| Dataset | Order1 | Order2 | Order3 | Order4 | Order5 | Average | Standard Deviation |
|---|---|---|---|---|---|---|---|
| ImageNet | 70.54 | 70.5 | 70.48 | 70.33 | 70.39 | 70.45 | 0.09 |
| ImageNet-A | 61.23 | 61.14 | 61.22 | 61.11 | 61.09 | 61.16 | 0.06 |
| ImageNet-V2 | 65.24 | 65.47 | 65.66 | 65.03 | 65.83 | 65.44 | 0.32 |
| ImageNet-R | 81.81 | 81.73 | 81.56 | 81.66 | 81.47 | 81.64 | 0.13 |
| ImageNet-S | 50.73 | 50.81 | 50.82 | 50.79 | 50.71 | 50.77 | 0.05 |

Table 16: Predictive cache logits with and without textual graph. "Logit Std" measures the standard deviation of logits across different cache capacities. "Max/Min Ratio" indicates the ratio between maximum and minimum logits

| Dataset | Method | Cache Capacity | | | | | |
|---|---|---|---|---|---|---|---|
| | | 0 | 1 | 2 | 3 | 4 | 5 |
| **ImageNet** | w/o textual graph | 0.0000 | 0.0006 | 0.0007 | 0.0008 | 0.0009 | 0.0011 |
| | w/ textual graph | 0.0007 | 0.0008 | 0.0009 | 0.0009 | 0.0010 | 0.0010 |
| **Caltech101** | w/o textual graph | 0.0000 | 0.0081 | 0.0090 | 0.0103 | 0.0118 | 0.0221 |
| | w/ textual graph | 0.0064 | 0.0093 | 0.0095 | 0.0101 | 0.0109 | 0.0142 |
| **Flowers102** | w/o textual graph | 0.0000 | 0.0060 | 0.0074 | 0.0106 | 0.0154 | 0.0211 |
| | w/ textual graph | 0.0073 | 0.0082 | 0.0095 | 0.0096 | 0.0101 | 0.0132 |
| **Stanford Cars** | w/o textual graph | 0.0000 | 0.0029 | 0.0042 | 0.0039 | 0.0046 | 0.0054 |
| | w/ textual graph | 0.0039 | 0.0041 | 0.0045 | 0.0044 | 0.0049 | 0.0052 |

