# OpenReview forum: "Rose-TTA: Reliable Online Structural Enhancement for Test-Time Adaptation"
_ICLR.cc/2026/Conference — ICLR 2026 Conference Desk Rejected Submission_

### Official Review · Reviewer_7oFc · 2025-10-17

**Soundness:** 3
**Presentation:** 3
**Contribution:** 2
**Rating:** 6
**Confidence:** 3

**Summary:**

Existing cache-based test-time adaption algorithms face several challenges, including unreliable cache by selecting test samples via entropy minimization, class imbalance in cache caused by random test example and insufficient sample-specific information for next test example. To address these issues, this paper proposes ROSE-TTA, a reliable online structural enhancement for test-time adaptation, by integrating several new techniques, including a noise-aware uncertainty measure for selecting cache entries, a graph-based structural completion strategy for mitigating class imbalance and completes global information and a sample-specific refinement mechanism for incorporating local information of each test example. The effectiveness of ROSE-TTA is verified by extensive experiments.

**Strengths:**

* This paper clearly explains the limitations of existing cache-based test-time adaption algorithms.

* This paper proposes several new techniques for cache-based test-time adaption algorithms which might be of independent interest.

**Weaknesses:**

I mainly concern the experimental results which are not sufficient.

* There is a lack of comparison on the cache accuracy of the proposed cache method on different datasets. Specifically, it is better to show the experimental results similar to Figure (a) and Figure (b), and compare it with the cache accuracy of existing methods. Similarly, the category and accuracy statistics (i.e., Figure (c), Figure (d)) of the proposed cache method should also be given and compared with existing cache methods.

* In section 2.2, the authors point out that the online test examples often arrive in random order in practice, naturally introducing class imbalance. Thus it is better to verify the robustness of the proposed algorithm to the random permutation of test examples. For instance, we can permute the test examples $N$ times, and report the average prediction performance and the standard derivation.

* There is a lack of comparison on the testing time. In the inference phase, the proposed algorithm computes $\hat{G}$, $P_{text}$ and $P_{image}$ which requires multiple matrix multiplication operations. It is better to verify the efficiency of the proposed algorithm with existing algorithms.

**Questions:**

Are the two conditions in (5) consistent? If not, it is possible that the two conditions can not be satisfied simultaneously. In this case, (5) is invalid.

---

> ### Author Response · Authors · 2025-11-21
>
> We thank Reviewer PpY7 for the insightful comments and valuable feedback. We have carefully addressed each of the concerns as follows:
> > **W1: There is a lack of comparison on the cache accuracy of the proposed cache method on different datasets… Specifically, it is better to show the experimental results similar to Figure (a) and Figure (b)… Similarly, the category and accuracy statistics (i.e., Figure (c), Figure (d))…**
>
>
> **Response:** **Comparisons on cache accuracy and biased prediction**
>
> To further demonstrate the effectiveness of the method, following your suggestions, we have now included additional comparisons of cache accuracy for multiple datasets, as illustrated in **Figure 7 in Appendix C.2** of the revised version. **These results demonstrate that our proposed ROSE-TTA method consistently outperforms existing cache-based methods, particularly in scenarios involving class imbalance.**
>
> Regarding the class imbalance problem in Figures (c) and (d), we would like to clarify that this phenomenon is **inherent to the nature of cache-based approaches and is primarily influenced by the input sample sequence, which cannot be explicitly mitigated.** In online test-time adaptation, test samples arrive sequentially in a random order. During the early adaptation stage, this random arrival leads to unequal cache capacities across classes.
>
>
> Our method **alleviate the biased cache logits caused by the class imbalance, rather than directly alleviating the class imbalance problem.** To further demonstrate this, we also provide an ablation study on the predictive cache logits with and without the textual graph. We average  all cache logits of categories with each cache capacity. As shown in the following table, without the textual graph, classes with fewer samples in the cache contribute very small logits (even 0). This leads to the early inference process being heavily influenced by a few dominant classes. Additionally, classes that have not yet appeared in the cache fail to provide any useful information, exacerbating the class imbalance. By contrast, with the textual graph, our method alleviates the biasBy contrast, with the textual graph, our method alleviates the bias with more balancing predictive logits. We added this experiment and discussions in Appendix C.8.
>
>
> | Dataset | | Cache Capacity=0 | Cache Capacity=1 | Cache Capacity=2 | Cache Capacity=3 | Cache Capacity=4 | Cache Capacity=5 |
> |---------|-------------|-------------|-------------|-------------|-------------|-------------|-------------|
> | Imagenet | w/o textual graph | 0.0000 | 0.0006 | 0.0007 | 0.0008 | 0.0009 | 0.0011 |
> | | w/ textual graph | 0.0007 | 0.0008 | 0.0009 | 0.0009 | 0.0010 | 0.0010 |
> | Caltech101 | w/o textual graph | 0.0000 | 0.0081 | 0.0090 | 0.0103 | 0.0118 | 0.0221 |
> | | w/ textual graph | 0.0064 | 0.0093 | 0.0095 | 0.0101 | 0.0109 | 0.0142 |
> | oxford_flowers | w/o textual graph | 0.0000 | 0.0060 | 0.0074 | 0.0106 | 0.0154 | 0.0211 |
> | | w/ textual graph | 0.0073 | 0.0082 | 0.0095 | 0.0096 | 0.0101 | 0.0132 |
> | stanford_cars | w/o textual graph | 0.0000 | 0.0029 | 0.0042 | 0.0039 | 0.0046 | 0.0054 |
> | | w/ textual graph | 0.0039 | 0.0041 | 0.0045 | 0.0044 | 0.0049 | 0.0052 |
> > **W2: In section 2.2, the authors point out that the online test examples often arrive in random order… Thus it is better to verify the robustness of the proposed algorithm to random permutation…  permute the test examples multiple times and report average performance and standard deviation.**
>
>
> **Response:** To show the robustness of our method regarding the test sample orders, we conducted multiple experiments on the OOD dataset with different sample orders and recorded the average prediction performance and standard deviation for each run.We have added a visualization of performance under different random seeds in **Figure 6 of the revised version (Section 5.2)**. The experimental results below demonstrate that **our method performs stably under random order conditions, confirming the robustness of the algorithm to the random permutation of test examples.**
>
> | Dataset     | Order1 | Order2 | Order3 | Order4 | Order5 | Average | Standard Deviation |
> |-------------|--------|--------|--------|--------|--------|---------|--------------------|
> | ImageNet    |  70.54 |   70.50 |  70.48 |  70.33 |  70.39 |   70.45 |               0.09 |
> | ImageNet-A  |  61.23 |  61.14 |  61.22 |  61.11 |  61.09 |   61.16 |               0.06 |
> | ImageNet-V2 |  65.24 |  65.47 |  65.66 |  65.03 |  65.83 |   65.44 |               0.32 |
> | ImageNet-R  |  81.81 |  81.73 |  81.56 |  81.66 |  81.47 |   81.64 |               0.13 |
> | ImageNet-S  |  50.73 |  50.81 |  50.82 |  50.79 |  50.71 |   50.77 |               0.05 |

---

> ### Author Response · Authors · 2025-11-21
>
> > **W3: There is a lack of comparison on the testing time… In the inference phase, the proposed algorithm computes $\hat{G}$, $P_{image}$, and $P_{text}$… which requires multiple matrix multiplications… It is better to verify efficiency with existing algorithms.**
>
>
> **Response:** We acknowledge that our method involves multiple matrix multiplication operations ($\hat{G}$, $P_{image}$, $P_{text}$​) during inference, which could potentially introduce overhead. We provide comprehensive testing time comparisons and component-level analysis to verify our efficiency claim. We have validated the efficiency from  experimental results.  We added this experiment and discussions in Appendix B and Section 5.2.
>
>
> | Method   | Testing Time(min) | Memory Usage(MB) | Accuracy(%) | Gain(%) |
> |----------|-------------------|------------------|----------|------|
> | CLIP     | 9.36             |           808.96 |    68.34 | _    |
> | TPT      | 585.00         |          4392.96 |    68.98 | 0.64 |
> | DiffTPT  | 1272.00         |           4710.40 |     70.30 | 1.96 |
> | TDA      | 50.00               |           860.16 |    69.51 | 1.17 |
> | ROSE-TTA |             48.96 |          1234.32 |    70.54 |  2.2 |
> - Our method achieves **26× faster** inference than DiffTPT while delivering **superior accuracy (+0.24%).** Compared to TPT, we are **12× faster** with **+1.56% accuracy gain**. This dramatic speedup stems from our training-free design—we avoid the expensive backward passes and prompt optimization required by test-time training methods. While TDA maintains **two separate caches (positive and negative)** to filter samples, ROSE-TTA achieves both **performance gains (+1.03%) and slight speedup (1.02×)** using only **a single cache with noise-aware quality assessment.
>
> To address the reviewer's specific concern about matrix multiplication overhead, we provide a detailed breakdown of testing time:
>
> | Component          | Testing Time(min) | Memory Usage(MB) |
> |--------------------|-------------------|------------------|
> | Noise Augmentation |              1.57 |           209.27 |
> | Graph Construction |              5.82 |           310.01 |
> | Sample Specific    |              1.41 |            22.89 |
> | Overall            |             48.96 |          1234.33 |
>
>
>
> The reviewer correctly identifies that our method involves multiple matrix multiplications: However, these operations contribute only **5.82 minutes (11.9% overhead)** because all matrix multiplications are embarrassingly parallel operations that fully exploit GPU tensor cores. All of our operations and matrix multiplications are conducted on the output features of CLIP model, whose computational costs are much fewer than that of the feedforward pass within the backbone.
>
>
> > **Q1: Are the two conditions in (5) consistent? If not, it is possible that the two conditions can not be satisfied simultaneously. In this case, (5) is invalid.**
>
> **Response:** No they are not consistent. In other words, it is possible that the two conditions are not satisfied simultaneously, which means the new coming test sample is not better than those stored in the cache. In this case, we will keep the original samples in the cache without any updating. We only update the cache when the two conditions are both satisfied.
>
> ---
>
> Please let us know if you have further questions -- thank you so much!

---

> > ### Comment · Reviewer_7oFc · 2025-11-23
> > **I thank the authors for detailed rebuttal that have addressed all of my concerns.**
> >
> > I will determine whether to maintain or raise the score after the discussions with other reviewers.

---

> ### Author Response · Authors · 2025-11-23
> **We are glad that our rebuttal have addressed all of your concerns ; )**
>
> Dear Reviewer 7oFc,
>
> We are glad that our rebuttal have addressed all of your concerns! Thank you so much for standing on the acceptance side of our work, and we are truly grateful for your dedicated time and constructive comments!
>
> Many thanks,
>
> Authors of #17055

---

### Official Review · Reviewer_PpY7 · 2025-10-31

**Soundness:** 2
**Presentation:** 2
**Contribution:** 2
**Rating:** 4
**Confidence:** 4

**Summary:**

The paper targets the limitations of cache-based test-time adaptation (TTA) for vision-language models under real-world distribution shifts. It introduces a noise-enhanced uncertainty criterion that combines entropy with prediction stability under noise, so the cache selects samples that are both low-entropy and robust to perturbations. To enable reliable cache utilization on unpredictable test streams, the method builds a graph modeling relationships between text embeddings and cached features, and performs sample-specific refinement to incorporate not only class-level but also instance-level information into the final prediction. Across 15 datasets, ROSE-TTA is compared against computationally heavy TPT-style baselines and lightweight cache-based TTA methods, showing good average performance.

**Strengths:**

- Compared to prior cache-based TTA, the method appears more stable and practical, demonstrated by rapid early-stage accuracy gains (Figure. 5) and noise-robust cache selection.

- The idea of leveraging Gaussian noise to measure prediction stability and improve cache reliability is novel and well-motivated.

**Weaknesses:**

- Narrow problem framing and incremental contribution. Improving the limitations of cache-based TTA feels like a small-scope problem rather than a new or consequential one, so the contribution reads as incremental rather than foundational.

- Modest and inconsistent empirical gains. Although the paper compares against two baseline families (TPT and prior cache-based methods) on 15 benchmarks, the improvements are not uniform across datasets and are often small in magnitude, so the overall effect feels weak.

- Missing analysis of cache dynamics and system overhead. Since the method centers on cache selection and utilization, please quantify replacement frequency over progress, cache hit ratio, and eviction reasons, as well as latency and memory usage from noise augmentation, graph construction, and sample-specific refinement. In particular, the noise augmentation design appears to require multiple forward passes per test sample, for example n=1 to n=20, which likely increases per-sample latency and compute cost. A table similar to [r1] Table 3 that reports accuracy together with runtime and memory under varying n would enable a fair comparison.

- Noise analysis does not make the gains visible. In Figure 4, varying the number of noise-augmented features from n=1 to 20 does not show a clear improvement, and most bars overlap within error ranges, so the claimed robustness gain is not evident. Including an n=0 no-noise baseline and reporting per-dataset deltas from n=0 would likely make the contribution of noise-based stability clearer and more compelling.

References:
[r1] Zhou, L., Ye, M., Li, S., Li, N., Zhu, X., Deng, L., Liu, H., & Lei, Z. (2025). Bayesian Test-Time Adaptation for Vision-Language Models. In Proceedings of the IEEE/CVF Conference on Computer Vision and Pattern Recognition (CVPR 2025).

**Questions:**

Q1. Figure 5 plots progress curves versus TDA, but Tables 1 and 2 show cases where BCA outperforms TDA. Why not include BCA in the progress comparison as well? If TDA was chosen intentionally, please clarify the rationale.

Q2. For cache methods, replacement frequency itself can introduce overhead. Could you report how often cache replacement occurs over progress, and compare noise-aware selection against entropy-only in terms of replacement rate and the associated time/memory overhead? This matters especially in real-world deployments where the update frequency of the cache directly impacts latency and compute cost.

Q3. In Table 3, the criteria for dataset selection are unclear. What was the selection rule? For fairness, would you report averages separately for the OOD benchmarks and for the cross-dataset suite rather than mixing them?

Typo. Tables 1 and 2 appear to mix the terms ROSE-TTA and ROSE-TDA. Which is the correct name? Please standardize the terminology throughout the paper.

---

> ### Author Response · Authors · 2025-11-21
>
> We thank Reviewer PpY7 for the insightful comments and valuable feedback. We have carefully addressed each of the concerns as follows:
> > **W1: Narrow problem framing and incremental contribution. Improving the limitations of cache-based TTA feels like a small-scope problem rather than a new or consequential one, so the contribution reads as incremental rather than foundational.**
>
> - **Response:** Thank you for your comments. While we acknowledge that cache-based TTA is becoming a well-established research direction, we still feel it's necessary to further clarify the novelty and significance of our work as follows.
>
> - **Why This Problem is Consequential**
>     -  Cache-based TTA is **the de facto standard for efficient deployment of VLMs** in production,  and it enables large pretrained models to better adapt to real-world applications. Our identified issues of **unreliable construction and incomplete utilization** affect every cache-based method.
>
> - **The Novelty of This Work**
>
>    - ROSE-TTA is the **first unified framework that addresses both cache construction and utilization reliability, co-designing the entire cache lifecycle.** Prior work (TDA, DOTA, BCA) primarily focuses on either construction or utilization, treating them as separate problems.  We recognize these are inherently coupled: unreliable cache construction cascades into biased utilization, while incomplete utilization fails to leverage even high-quality cached samples.
>
>   - Our noise-enhanced stability metric (Eq. 2) introduces a fundamentally new principle: prediction robustness under perturbations as a complementary signal to confidence, which forces the cache features both distinguishable and stable, enhancing their reliability.
>
>   - Our tri-graph reconstruction  is the first to **leverage text embeddings' complete semantic structure to compensate for underrepresented classes and  weight cache samples by reliability** to prevent noisy samples from corrupting the graph.
>
>   - We are the first work in TTA to use **pseudo-gradient formulation  for mimicking entropy minimization's directionality** without backpropagation.
>
> - We believe these contributions significantly advance the field of test-time adaptation for VLMs and establish new principles for future research. We hope the added theoretical analysis and empirical validation (detailed above) adequately address the reviewer's concerns.

---

> ### Author Response · Authors · 2025-11-21
>
> > **W2: Modest and inconsistent empirical gains. Although the paper compares against two baseline families on 15 benchmarks, the improvements are not uniform across datasets and are often small in magnitude, so the overall effect feels weak.**
>
> **Response:** Thank you for your comments. We have compared our method with both fine-tuning-based method and efficient adaptation methods on 15 benchmarks and achieves first or second rank on most of them, achieving overall the best performance.
>
> To further show the effectiveness of our method, we have incorporated experiments on OOD corruption datasets such as **CIFAR-10-C, CIFAR-100-C, and ImageNet-C** with CLIP-ViT-B/16 as our backbone. We reproduced zero-shot CLIP , TDA and BCA on the datasets for comparisons. The results are in the following tables. **Our ROSE-TTA achieves better overall performance and consistent empirical gains on these three datasets compared with CLIP , TDA and BCA, demonstrating the effectiveness of our method on these challenging distribution shifts.**   We added this experiment and discussions in Appendix C.1.
>
> |  | Guassian | Shot | Impulse | Defocus | Glass | Motion | Zoom | Snow | Frost | Fog | Brightness | Contrast | Elastic | Pixelate | JPEG | Mean |
> |--------|-----------|------|---------|---------|-------|--------|------|------|-------|-----|-----------|---------|---------|----------|------|------|
> | **ImageNet-C** |
> | CLIP | 18.56 | 12.11 | 19.45 | 31.22 | 23.12 | 32.15 | 26.25 | 34.59 | 34.17 | 45.35 | 57.11 | 30.94 | 48.93 | 45.01 | 35.14 | 32.94 |
> | TDA | 19.54 | 13.46 | 19.00 | 32.76 | 26.35 | 34.32 | 27.79 | 36.42 | 35.47 | 47.79 | 58.16 | 31.92 | 49.22 | 45.05 | 35.80 | 34.26 |
> | BCA | 19.76 | 13.44 | 20.38 | 33.08 | 27.84 | 34.62 | 27.88 | 38.12 | 35.96 | 52.26 | 59.02 | 32.38 | 49.20 | 45.08 | 35.94 | 34.99 |
> | Ours | **20.18** | **13.57** | **21.56** | **33.67** | **29.78** | **35.19** | **28.12** | **40.17** | **36.66** | **60.89** | **59.94** | **33.26** | **49.38** | **45.18** | **36.21** | **36.25** |
> | **CIFAR-10C** |
> | CLIP| 46.22 | 50.17 | 45.23 | 65.45 | 43.33 | 60.14 | 68.54 | 69.12 | 71.11 | 55.23 | 80.45 | 44.53 | 55.25 | 52.17 | 60.31 | 57.82 |
> | TDA | 48.66 | 53.24 | 46.26 | 67.09 | 43.55 | 63.17 | 69.97 | 70.02 | 72.57 | 56.13 | 81.90 | 45.59 | 56.84 | 53.71 | 60.91 | 59.31|
> | BCA | 49.08 | 53.72 | 46.18 | 68.00 | 44.55 | 65.92 | 70.82 | 71.38 | 72.98 | 56.75 | 81.98 | 47.05 | 56.35 | 54.02 | 61.52 | 60.03 |
> | Ours | **50.14** | **54.76** | **46.31** | **69.73** | **46.17** | **69.56** | **72.45** | **73.52** | **74.23** |**57.86** | **82.04** | **49.23** | **56.97** | **54.75** | **62.69** | **61.36**|
> | **CIFAR-100C**  |
> | CLIP | 28.15 | 28.18 | 20.17 | 38.22 | 20.85 | 35.24 | 42.56 | 40.14 | 42.88 | 30.13 | 52.19 | 20.89 | 29.66 | 24.15 | 33.11 | 32.43 |
> | TDA | 29.18 | 29.84 | 24.68 | 39.44 | 20.99 | 37.44 | 43.84 |42.86 | 44.24 | 30.72 | 53.72 | 22.85 | 30.78 | 25.13 | 33.81 | 33.97 |
> | BCA | 29.66 | 30.10 | 25.96 | 40.36 | 21.64 | 37.70 | 43.62 | 42.92 | 46.28 | 30.79 | 54.90 | 23.74 | 32.16 | 25.18 | 33.88 | 34.51 |
> | Ours | **30.22** | **30.46** | **27.89** | **41.58** | **22.45** | **38.11** | **44.19** | **43.00** | **48.69**| **30.87** | **56.24** | **24.78** | **33.94** | **25.22** | **33.96** | **35.44** |

---

> ### Author Response · Authors · 2025-11-21
>
> > **W3: Missing analysis of cache dynamics and system overhead… quantify replacement frequency, cache hit ratio, eviction reasons… report latency and memory from noise augmentation, graph construction, sample-specific refinement… multiple forward passes (n=1–20) likely increase cost… a table with accuracy, runtime, memory under varying n would allow fair comparison.**
>
>
> **Response:** Thank you for your valuable suggestion! We agree that a thorough analysis of cache dynamics and computational overhead is essential for understanding our method's practical behavior. We follow your suggestion to **add a detailed analysis of cache dynamics and system overhead**, demonstrating the practical efficiency of our method.
>
> **1. Cache Dynamics Analysis**
> - **Cache Replacement Statistics Across All 15 Benchmarks**
>   Across all 15 datasets, the average replacement ratio is only **12.7%**, meaning  cached samples **remain stable throughout testing**. This indicates our **noise-aware selection effectively identifies truly representative prototypes.**  We added this experiment and discussions in Appendix C.6.
>
>
> |                  | ImageNet | ImageNet-A  | ImageNet-V2 | ImageNet-R | ImageNet-S |
> |------------------|----------|-------------|-------------|------------|------------|
> | **TDA**          |          |             |             |            |            |
> | Replacement Num  |    15762 |        1456 |        2384 |       7825 |      13184 |
> | Replacement Ratio|     0.31 |        0.19 |        0.24 |       0.27 |       0.26 |
> | **Ours**         |          |             |             |            |            |
> | Replacement Num  |     8846 |         801 |        1325 |       4375 |       7346 |
> | Replacement Ratio|     0.18 |        0.11 |        0.13 |       0.15 |       0.15 |
>
>
> |                  | Aircraft | Caltech101 | Cars  | DTD  | EuroSAT | Flower102 | Food101 | Pets | SUN397 | UCF101 |
> |------------------|----------|------------|-------|------|---------|-----------|---------|------|--------|--------|
> | **TDA**          |          |            |       |      |         |           |         |      |        |        |
> | Replacement Num  |      528 |        443 |  1339 |  258 |     118 |       513 |    8546 |  314 |   2178 |    236 |
> | Replacement Ratio|     0.16 |        0.18 |  0.16 | 0.14 |    0.02 |      0.29 |    0.28 | 0.09 |   0.11 |   0.05 |
> | **Ours**         |          |            |       |      |         |           |         |      |        |        |
> | Replacement Num  |      300 |        246 |   744 |  143 |      65 |       285 |    4753 |  174 |   1211 |    131 |
> | Replacement Ratio|     0.09 |        0.10 |  0.09 | 0.08 |    0.01 |      0.16 |    0.16 | 0.05 |   0.06 |   0.03 |
>
> - **Clarification on "Cache Hit Ratio":**
>     We respectfully note that the traditional concept of "cache hit ratio" does not directly apply to our method due to fundamental differences in cache design philosophy. Unlike conventional caching systems (e.g., CPU caches, key-value stores) where a "hit" means finding a requested item and a "miss" requires fetching from a slower source, **our cache serves a different purpose of Knowledge propagation via prototypes**.
>
>
> - **Clarification on "Eviction Reasons":** ROSE-TTA employs **a single, unified eviction strategy** based on noise-aware quality scoring. There are **no "multiple eviction reasons"** or conditional replacement path. To validate our eviction design, we conducted ablation studies comparing ROSE-TTA's noise-aware joint scoring against common cache replacement policies: (1) Entropy-only, (2) Similarity, (3) Random, (4) FIFO (First-In-First-Out), (5) LRU (Least Recently Used). **Our noise-aware joint score consistently outperforms all alternatives**, achieving +0.81% on cross-dataset and +0.91% on OOD compared to the best baseline (entropy-only). This validates the importance of combining both confidence and robustness. We added this experiment and discussions in Appendix C.7.
>
>
> | Method          | Cross Datset Avg | OOD Avg |
> |--------------|------------------|---------|
> | Entropy only   |            68.21 |     65.13 |
> | Similarity     |            68.13 |      64.99 |
> | Random         |            67.11 |      63.64 |
> | FIFO           |            64.68 |      62.25 |
> | LRU             |            63.21 |      60.02 |
> | Ours            |            69.20 |     66.04 |

---

> ### Author Response · Authors · 2025-11-21
>
> **2. System Overhead Breakdown**
>
>  We have validated the efficiency from  experimental results. We added this experiment and discussions in Appendix B and Section 5.2.
>
> - **Overall Performance Analysis:** Table below presents a comprehensive comparison of ROSE-TTA against baseline CLIP and state-of-the-art test-time adaptation methods on ImageNet .Our method achieves **26× faster** inference than DiffTPT while delivering **superior accuracy (+0.24%).** Compared to TPT, we are **12× faster** with **+1.56% accuracy gain**. This dramatic speedup stems from our training-free design—we avoid the expensive backward passes and prompt optimization required by test-time training methods. While TDA maintains **two separate caches (positive and negative)** to filter samples, ROSE-TTA achieves both **performance gains (+1.03%) and slight speedup (1.02×)** using only **a single cache with noise-aware quality assessment.**
>
>
> | Method   | Testing Time(min) | Memory Usage(MB) | Accuracy(%) | Gain(%) |
> |----------|-------------------|------------------|----------|------|
> | CLIP     | 9.36             |           808.96 |    68.34 | _    |
> | TPT      | 585.00         |          4392.96 |    68.98 | 0.64 |
> | DiffTPT  | 1272.00         |           4710.40 |     70.30 | 1.96 |
> | TDA      | 50.00               |           860.16 |    69.51 | 1.17 |
> | ROSE-TTA |             48.96 |          1234.32 |    70.54 |  2.20 |
>
> - **Component Analysis:** The table below breaks down the computational cost of each component:
>
> | Component          | Testing Time(min) | Memory Usage(MB) |
> |--------------------|-------------------|------------------|
> | Noise Augmentation |              1.57 |           209.27 |
> | Graph Construction |              5.82 |           310.01 |
> | Sample Specific    |              1.41 |            22.89 |
> | Overall            |             48.96 |          1234.33 |
>
>
>
> (1) Graph Construction:
>   - Though graph construction involves a large amount of  matrix multiplications, **All these operations are fully parallelizable across GPU cores with no sequential dependencies.**
>
> (2) Noise Augmentation:
>   - Noise vectors are **pre-generated globally once at initialization** Per-sample operations involve only lightweight vector addition (broadcasting) and batched matrix multiplication
>
>   - Every test sample uses the same **pre-computed noise bank** via simple broadcasting. There is no per-sample noise generation overhead.
>
> (3) Sample-Specific Adaptation:
>   - By **avoiding true gradient computation** , we eliminate the dominant computational cost of test-time training methods while still achieving dynamic, sample-specific adaptation.
>
>
>
> **3. Scalability with Varying Noise Budgets (ImageNet)**
>
> Testing across 50,000 ImageNet samples shows that  our noise augmentation design **does not require multiple forward passes per test sample**. That's because we pre-generate n noise patterns offline and apply them efficiently during inference through cached operations.  It demonstrates that our method is computationally efficient and practical for deployment, as the per-sample overhead remains minimal even with large noise budgets. We added this experiment and discussions in Appendix B.2.
>
> | Noise Num | Testing Time (ms/sample) | Memory Usage (KB/sample) |
> |-----------|--------------------------|--------------------------|
> |         1 |                    58.36 |                    21.40 |
> |         3 |                    58.51 |                    22.29 |
> |         5 |                    58.68 |                    23.17 |
> |        10 |                    58.75 |                    25.28 |
> |        15 |                    59.17 |                    27.53 |
> |        20 |                    59.69 |                    29.64 |

---

> ### Author Response · Authors · 2025-11-21
>
> > **W4: Noise analysis does not make the gains visible. In Figure 4, varying the number of noise-augmented features from n=1 to 20 does not show a clear improvement, and most bars overlap within error ranges, so the claimed robustness gain is not evident. Including an n=0 no-noise baseline and reporting per-dataset deltas from n=0 would likely make the contribution of noise-based stability clearer and more compelling.**
>
>
> **Response:** Following your suggestion, we have added the comparisons with the no-noise-augmentation baseline (n=0) to more clearly demonstrate the benefits of noise augmentation. **We added the baseline of *n = 0* in Figure 5 for a clear comparison.** Below, we present the performance of each dataset at different noise levels, where **"Noise Gains"** represents the average performance gain relative to the no-noise baseline. As shown, **noise augmentation obviously improves classification performance** in most cases, and **our method exhibits good stability across different noise levels.** We added this experiment and discussions in Appendix C.5.
> |             | n = 0      | n = 1      |                | n = 3      |                | n = 5      |                | n = 10     |                | n = 15     |                | n = 20     |                |
> |-------------|------------|------------|----------------|------------|----------------|------------|----------------|------------|----------------|------------|----------------|------------|----------------|
> |             | Accuracy   | Accuracy   | Noise Gain     | Accuracy   | Noise Gain     | Accuracy   | Noise Gain     | Accuracy   | Noise Gain     | Accuracy   | Noise Gain     | Accuracy   | Noise Gain     |
> | Aircraft    | 23.51±0.41 | 24.78±0.82 | **1.27±0.23**  | 25.14±0.77 | **1.63±0.24**  | 25.16±0.12 | **1.65±0.11**  | 25.86±0.14 | **2.35±0.11**  | 25.14±0.79 | **1.63±0.22**  | 25.11±0.78 | **1.60±0.25**  |
> | Caltech101  | 94.16±0.01 | 94.52±0.03 | **0.36±0.01**  | 94.62±0.05 | **0.46±0.02**  | 94.63±0.01 | **0.47±0.01**  | 94.76±0.05 | **0.60±0.03**  | 94.61±0.02 | **0.45±0.01**  | 94.59±0.02 | **0.43±0.01**  |
> | Cars        | 65.73±0.23 | 66.78±0.76 | **1.05±0.33**  | 66.27±0.74 | **0.54±0.43**  | 66.61±0.66 | **0.88±0.56**  | 67.06±0.54 | **1.33±0.64**  | 66.57±0.45 | **0.84±0.35**  | 66.91±0.44 | **1.18±0.52**  |
> | DTD         | 44.61±0.39 | 47.34±0.76 | **2.73±0.63**  | 47.38±0.89 | **2.77±0.77**  | 47.56±0.98 | **2.95±0.82**  | 48.00±0.78 | **3.39±0.67**  | 47.64±0.33 | **3.03±0.23**  | 47.79±0.56 | **3.18±0.55**  |
> | EuroSAT     | 58.67±0.24 | 64.25±1.01 | **5.58±0.78**  | 64.86±0.74 | **6.19±0.69**  | 64.34±0.56 | **5.67±0.73**  | 65.64±0.62 | **6.97±0.54**  | 65.12±0.29 | **6.45±0.13**  | 65.32±0.27 | **6.65±0.09**  |
> | Flower102   | 71.41±0.23 | 73.37±1.45 | **1.96±0.79**  | 73.89±1.32 | **2.48±1.23**  | 73.65±1.57 | **2.24±1.21**  | 74.17±1.02 | **2.76±0.98**  | 73.44±1.23 | **2.03±1.01**  | 72.71±0.99 | **1.30±0.45**  |
> | Food101     | 86.11±0.01 | 86.18±0.00 | **0.07±0.01**  | 86.17±0.01 | **0.06±0.01**  | 86.15±0.03 | **0.04±0.02**  | 86.21±0.00 | **0.10±0.01**  | 86.2±0.01  | **0.09±0.01**  | 86.19±0.03 | **0.08±0.02**  |
> | Pets        | 89.17±0.43 | 89.72±0.89 | **0.55±0.42**  | 90.87±0.67 | **1.70±0.78**  | 90.32±0.79 | **1.15±0.72**  | 90.95±1.01 | **1.78±0.92**  | 89.92±0.65 | **0.75±0.52**  | 89.97±0.78 | **0.80±0.62**  |
> | SUN397      | 66.62±0.12 | 67.88±0.13 | **1.26±0.08**  | 67.76±0.24 | **1.14±0.15**  | 67.54±0.03 | **0.92±0.07**  | 68.00±0.15 | **1.38±0.14**  | 67.36±0.11 | **0.74±0.08**  | 67.68±0.15 | **1.06±0.21**  |
> | UCF101      | 68.12±0.25 | 70.23±0.67 | **2.11±0.42**  | 70.68±0.56 | **2.56±0.55**  | 71.29±0.78 | **3.17±0.74**  | 71.34±0.89 | **3.22±0.92**  | 71.11±0.85 | **2.99±0.93**  | 70.98±1.01 | **2.86±0.97**  |
> | ImageNet    | 70.05±0.01 | 70.24±0.04 | **0.19±0.04**  | 70.48±0.07 | **0.43±0.06**  | 70.29±0.04 | **0.24±0.03**  | 70.54±0.06 | **0.49±0.05**  | 70.31±0.05 | **0.26±0.05**  | 70.44±0.04 | **0.39±0.03**  |
> | ImageNet-A  | 59.43±0.06 | 60.87±0.06 | **1.44±0.01**  | 61.15±0.02 | **1.72±0.03**  | 60.96±0.03 | **1.53±0.03**  | 61.22±0.05 | **1.79±0.06**  | 61.21±0.02 | **1.78±0.05**  | 61.03±0.04 | **1.60±0.03**  |
> | ImageNet-V2 | 64.35±0.21 | 65.11±0.25 | **0.76±0.11**  | 65.25±0.31 | **0.90±0.31**  | 65.76±0.09 | **1.41±0.15**  | 65.83±0.32 | **1.48±0.25**  | 65.81±0.13 | **1.46±0.11**  | 65.72±0.23 | **1.37±0.22**  |
> | ImageNet-R  | 80.37±0.13 | 81.54±0.11 | **1.17±0.04**  | 81.64±0.13 | **1.27±0.05**  | 81.79±0.45 | **1.42±0.25**  | 81.81±0.03 | **1.44±0.14**  | 81.77±0.14 | **1.40±0.13**  | 81.57±0.09 | **1.20±0.11**  |
> | ImageNet-S  | 49.41±0.02 | 50.67±0.05 | **1.26±0.02**  | 50.74±0.05 | **1.33±0.03**  | 50.72±0.07 | **1.31±0.04**  | 50.82±0.07 | **1.41±0.05**  | 50.81±0.04 | **1.40±0.01**  | 50.69±0.03 | **1.28±0.02**  |

---

> ### Author Response · Authors · 2025-11-21
>
> > **Q1:Figure 5 plots progress curves versus TDA, but Tables 1 and 2 show cases where BCA outperforms TDA. Why not include BCA in the progress comparison as well? If TDA was chosen intentionally, please clarify the rationale.**
>
> **Response:** We chose to include TDA in the progress comparison due to its relevance to our method, as it is a typical cache-based baseline.  By focusing on TDA, we can better demonstrate how our uncertainty-aware cache construction addresses the early-stage performance collapse issue, which is often observed in methods relying solely on entropy for selection.
> Following your suggestion, we have also conducted **a comparison between BCA and our method**, and the results are revised in the Fig.5 in revised version. The conclusion is similar. our approach still **performs larger improvements on early stages**, which demonstrates the migiation of biased prediction.
>
> > **Q2. For cache methods, replacement frequency itself can introduce overhead. Could you report how often cache replacement occurs over progress, and compare noise-aware selection against entropy-only in terms of replacement rate and the associated time/memory overhead? This matters especially in real-world deployments where the update frequency of the cache directly impacts latency and compute cost.**
>
>
>
> **Response:** We provide a comprehensive analysis of replacement frequency and its associated computational overhead. Our noise-aware selection achieves **43.9% fewer replacements than entropy-only** , while simultaneously delivering: **faster inference and higher accuracy**. The memory usage increases modestly by 13.6% due to the noise-augmentation structures. This demonstrates that the quality-driven selection strategy not only improves accuracy but also efficiency by minimizing unnecessary cache updates. We added this experiment and discussions in Appendix C.7.
>
> | Method       | Frequency | Testing Time(min) | Memory Usage(MB) |
> |--------------|-----------|-------------------|------------------|
> | Entropy only |     15762 |             49.88 |          1086.11 |
> | Noise aware  |      8846 |             48.96 |          1234.33 |
>
>
> > **Q3. In Table 3, the criteria for dataset selection are unclear. What was the selection rule? For fairness, would you report averages separately for the OOD benchmarks and for the cross-dataset suite rather than mixing them?**
>
> **Response:** Thank you for your valuable comment. To clarify, we randomly selected four cross-dataset samples for demonstrating the impact of each component in our model. In response to your suggestion, we have updated the results to report the averages separately for the OOD benchmarks and the cross-dataset suite to ensure fairness. The updated experimental results are presented in the table below, which clearly shows that **all three components contribute to improving the performance and stability of cache-based methods.** We have updated the Table 3 in the  revised version. We have revsied the Table 3.
>
> | Method | Cross Dataset Avg | OOD Avg |
> |--------|-------------------|---------|
> | w/o $\mathbf{d}$ | 67.89 | 65.03 |
> | w/o $P_{image}$ & $P_{text}$ | 68.24 | 65.22 |
> | w/o $\hat{F}$ | 68.78 | 65.68 |
> | **Rose-TTA** | 69.20 | 66.04 |
>
> > **Typo. Tables 1 and 2 appear to mix the terms ROSE-TTA and ROSE-TDA. Which is the correct name? Please standardize the terminology throughout the paper.**
>
> **Reponse:** The correct name is ROSE-TTA. We have fixed the typos throught our paper - thank you so much!
>
> ---
>
> Please let us know if we have properly addressed your questions and we are more than happy to discuss more!

---

> ### Comment · Reviewer_PpY7 · 2025-11-28
> **Thank you for the rebuttal.**
>
> Some of my concerns are addressed with your sincere rebuttal, but I keep my score because the contribution still feels incremental in scope, and the empirical gains, although strengthened by the additional OOD-corruption experiments, remain small in magnitude and not consistently strong across the original benchmarks.

---

### Official Review · Reviewer_rmg6 · 2025-11-01

**Soundness:** 3
**Presentation:** 2
**Contribution:** 2
**Rating:** 4
**Confidence:** 2

**Summary:**

ROSE-TTA is a training-free, online TTA method for CLIP that (1) keeps only reliable cache items (improves sample selection for cache), (2) completes class info with a class graph, and (3) does a light per-sample refinement. It shows some gains on cross-dataset/OOD tests, but not uniformly.

**Strengths:**

**s1: Originality**
Nice idea that combines noise-aware cache selection, class-graph completion, and per-sample refinement in a training-free, online setting.

**s2: Quality**
The paper includes multiple ablations that isolate the effect of each component, which makes the method much more convincing.

**s3: Practicality**
No backprop at test time, works with batch = 1, and seems like an efficient and deployable approach.

**Weaknesses:**

**w1: Consistency**
Improvements are not universal across all datasets, even though the method is usually competitive (often first or second).

**w2: Sensitivity** Results likely depend on cache size/noise/refinement hyperparameters.

**w3: Efficiency discussion**
The paper could more explicitly highlight and quantify its efficiency (runtime, memory) compared to other TTA baselines.

**Questions:**

**q1: Pseudo-gradient vs true gradient**
I am not fully sure how the pseudo-gradient you use compares to the actual gradient. Could you either prove this more explicitly or show empirical comparisons (e.g., against a small true-gradient baseline) to clarify how close it is?

**q2: Efficiency**
Since the method is training-free and online, could you report wall-clock time and memory usage versus other TTA methods, and maybe highlight efficiency as a main selling point in the discussion?

---

> ### Author Response · Authors · 2025-11-21
>
> We thank Reviewer rmg6 for the insightful comments and valuable feedback. We have carefully addressed each of the concerns as follows:
>
> > **W1: Consistency Improvements are not universal across all datasets, even though the method is usually competitive (often first or second).**
>
> **Response:** Different datasets may present varying levels of complexity in terms of **class distribution, feature representation, and noise**. In some cases, the method's strengths in **leveraging textual embeddings for class relationships may be more pronounced**, while in others, the model may face challenges due to these intrinsic differences.
>
> To further show the effectiveness of our method, we have incorporated experiments on OOD corruption datasets such as **CIFAR-10-C, CIFAR-100-C, and ImageNet-C** with CLIP-ViT-B/16 as our backbone. We reproduced zero-shot CLIP , TDA and BCA on the datasets for comparisons. The results are in the following tables. **Our ROSE-TTA achieves better overall performance and consistent empirical gains on these three datasets compared with CLIP , TDA and BCA, demonstrating the effectiveness of our method on these challenging distribution shifts.**  We added this experiment and discussions in Appendix C.1.
>
> |  | Guassian | Shot | Impulse | Defocus | Glass | Motion | Zoom | Snow | Frost | Fog | Brightness | Contrast | Elastic | Pixelate | JPEG | Mean |
> |--------|-----------|------|---------|---------|-------|--------|------|------|-------|-----|-----------|---------|---------|----------|------|------|
> | **ImageNet-C** |
> | CLIP | 18.56 | 12.11 | 19.45 | 31.22 | 23.12 | 32.15 | 26.25 | 34.59 | 34.17 | 45.35 | 57.11 | 30.94 | 48.93 | 45.01 | 35.14 | 32.94 |
> | TDA | 19.54 | 13.46 | 19.90 | 32.76 | 26.35 | 34.32 | 27.79 | 36.42 | 35.47 | 47.79 | 58.16 | 31.92 | 49.22 | 45.05 | 35.80 | 34.26 |
> | BCA | 19.76 | 13.44 | 20.38 | 33.08 | 27.84 | 34.62 | 27.88 | 38.12 | 35.96 | 52.26 | 59.02 | 32.38 | 49.20 | 45.08 | 35.94 | 34.99 |
> | Ours | **20.18** | **13.57** | **21.56** | **33.67** | **29.78** | **35.19** | **28.12** | **40.17** | **36.66** | **60.89** | **59.94** | **33.26** | **49.38** | **45.18** | **36.21** | **36.25** |
> | **CIFAR-10C** |
> | CLIP| 46.22 | 50.17 | 45.23 | 65.45 | 43.33 | 60.14 | 68.54 | 69.12 | 71.11 | 55.23 | 80.45 | 44.53 | 55.25 | 52.17 | 60.31 | 57.82 |
> | TDA | 48.66 | 53.24 | 46.26 | 67.09 | 43.55 | 63.17 | 69.97 | 70.02 | 72.57 | 56.13 | 81.90 | 45.59 | 56.84 | 53.71 | 60.91 | 59.31|
> | BCA | 49.08 | 53.72 | 46.18 | 68.00 | 44.55 | 65.92 | 70.82 | 71.38 | 72.98 | 56.75 | 81.98 | 47.05 | 56.35 | 54.02 | 61.52 | 60.03 |
> | Ours | **50.14** | **54.76** | **46.31** | **69.73** | **46.17** | **69.56** | **72.45** | **73.52** | **74.23** |**57.86** | **82.04** | **49.23** | **56.97** | **54.75** | **62.69** | **61.36**|
> | **CIFAR-100C**  |
> | CLIP | 28.15 | 28.18 | 20.17 | 38.22 | 20.85 | 35.24 | 42.56 | 40.14 | 42.88 | 30.13 | 52.19 | 20.89 | 29.66 | 24.15 | 33.11 | 32.43 |
> | TDA | 29.18 | 29.84 | 24.68 | 39.44 | 20.99 | 37.44 | 43.84 |42.86 | 44.24 | 30.72 | 53.72 | 22.85 | 30.78 | 25.13 | 33.81 | 33.97 |
> | BCA | 29.66 | 30.10 | 25.96 | 40.36 | 21.64 | 37.70 | 43.62 | 42.92 | 46.28 | 30.79 | 54.90 | 23.74 | 32.16 | 25.18 | 33.88 | 34.51 |
> | Ours | **30.22** | **30.46** | **27.89** | **41.58** | **22.45** | **38.11** | **44.19** | **43.00** | **48.69**| **30.87** | **56.24** | **24.78** | **33.94** | **25.22** | **33.96** | **35.44** |

---

> ### Author Response · Authors · 2025-11-21
>
> > **W2:  Sensitivity Results likely depend on cache size/noise/refinement hyperparameters.**
>
> **Response:** We understand that the sensitivity of our method might be influenced by cache size, noise, and refinement hyperparameters. Below is a more concise discussion:
>
> - **Sensitivity Results on Cache Size**: As shown in Figure 3, ROSE-TTA performs best with a cache size of 5, **maintaining stability on most datasets**.  On datasets like Flower102, larger cache sizes introduce redundant or noisy samples, causing instability due to highly similar classes.
>
>
> - **Sensitivity Results on Noise Augmentations**: We also investigated the effect of noise augmentations on stability estimation, with results showing slight performance improvement as augmentations increased, where we can usually get good performance at around 10 (see Figure 4), which is also **tend to be consistent**.
>
>
> - **Sensitivity Results on Refinement Hyperparameters**:Regarding the refinement hyperparameters, $\gamma$ and $\zeta$, these are **adaptive** and serve as **intensity control factors**. The pseudo-learning rate $\eta$ is **empirically and consistently set to 0.01.** We have clarified this in the Section 3.2.
>
>
> We have also  conducted additional experiments across all datasets to evaluate the performance under different values of $\alpha$. Our results show that the performance **remains relatively stable with changes in $\alpha$** and significantly outperforms the case where $\alpha=0$ (which degenerates to using only zero-shot CLIP predictions). Notably, across the majority of $\alpha$ values tested, our method  outperforms BCA on most datasets, demonstrating  effectiveness even with randomly selected   $\alpha$.   We added this experiment and discussions in Appendix C.3.
>
> | Dataset       | Aircraft | Caltech101 | Cars  | DTD   | EuroSAT | Flower102 | Food101 | Pets   | SUN397 | UCF101 | ImageNet | ImageNet-A | ImageNet-V2 | ImageNet-R | ImageNet-S | Mean   |
> |---------------|----------|------------|-------|-------|---------|-----------|---------|--------|--------|--------|----------|------------|-------------|------------|------------|--------|
> | $\alpha$ = 0  | 24.70    | 94.16      | 65.80 | 44.50 | 47.67   | 71.42     | 86.10   | 89.15  | 66.62  | 66.77  | 70.04    | 59.43      | 64.35       | 80.33      | 49.40      | 65.36  |
> | BCA | **28.59**    | 94.69      | 66.86 | 53.49 | 56.63   | 73.12     | 85.97 | 90.43  | **68.41** | 67.59  | 70.22    | 61.14     | 64.90   | 80.72      |**50.87**     | 67.58 |
> | $\alpha$ = 1  | 24.77    | 94.45      | 66.92 | 47.78 | 60.12   | 73.42     | 86.12   | **90.95** | 67.43  | **71.34** | 70.11    | 60.11      | 64.90       | 81.33      | 50.32      | 67.40  |
> | $\alpha$ = 3  | 25.86| 94.50      | **67.06** | **48.00** | **65.64** | 73.67     | 86.15   | 90.42  | 68.00 | 70.33  | **70.54** | **61.22**  | 65.68       | 81.56      | 50.45      | **67.94**  |
> | $\alpha$ = 5  | 25.23    | **94.76**  | 66.77 | 47.90 | 65.50   | **74.17** | 86.16   | 90.56  | 67.56  | 69.57  | 70.33    | 60.58      | 65.77       | **81.81**  | 50.82  | 67.83  |
> | $\alpha$ = 10 | 25.69    | 94.68      | 67.02 | 47.22 | 63.25   | 74.11     | **86.16** | 90.66  | 67.78  | 70.12  | 70.42    | 61.01      | **65.83**   | 81.23      | 50.64      | 67.73 |

---

> ### Author Response · Authors · 2025-11-21
>
> > **W3: The paper could more explicitly highlight and quantify its efficiency (runtime, memory) compared to other TTA baselines.**
>
> > **Q2: Efficiency Since the method is training-free and online, could you report wall-clock time and memory usage versus other TTA methods, and maybe highlight efficiency as a main selling point in the discussion?**
>
>
> **Response:** Thank you for this excellent suggestion. We agree that efficiency is indeed a crucial advantage of our method and appreciate the opportunity to highlight this aspect more prominently. We have validated the efficiency from  experimental results.
> We added this experiment and discussions in Section 5.2.
>
> We have conducted comprehensive efficiency benchmarks on ImageNet across representative TTA methods. Table below presents the comparison:
>
> | Method   | Testing Time(min) | Memory Usage(MB) | Accuracy(%) | Gain(%) |
> |----------|-------------------|------------------|----------|------|
> | CLIP     | 9.36             |           808.96 |    68.34 | _    |
> | TPT      | 585.00         |          4392.96 |    68.98 | 0.64 |
> | DiffTPT  | 1272.00         |           4710.40 |     70.30 | 1.96 |
> | TDA      | 50.00               |           860.16 |    69.51 | 1.17 |
> | ROSE-TTA |             48.96 |          1234.32 |    70.54 |  2.20 |
>
> Our method achieves **26× faster** inference than DiffTPT while delivering **superior accuracy (+0.24%).** Compared to TPT, we are **12× faster** with **+1.56% accuracy gain**. This dramatic speedup stems from our training-free design where we avoid the expensive backward passes and prompt optimization required by test-time training methods.  While TDA maintains two separate caches (positive and negative) to filter samples, ROSE-TTA achieves both **performance gains (+1.03%) and slight speedup (1.02×)** using only **a single cache with noise-aware quality assessment.**

---

> ### Author Response · Authors · 2025-11-21
>
> > **Q1: Pseudo-gradient vs true gradient. I am not fully sure how the pseudo-gradient you use compares to the actual gradient. Could you either prove this more explicitly or show empirical comparisons (e.g., against a small true-gradient baseline) to clarify how close it is?**
>
> **Response:**
> To compare the pseudo-gradient and actual gradient of entropy minimization, we provide both **theoretical analyses** and **empirical results** in the following. We added this experiment and discussions in Appendix A.
>
>
>
> The common gradient-based method for test-time adaptation calculates gradients based on **entropy minimization**
>
> $$
> H(p) = -\sum_{k=1}^C p_k \log p_k,
> $$
>
> where $p_k$ denotes the predicted probability for class $k$ and $C$ is the number of classes.
>
> In our **sample-specific cache refinement** **(Eq. 9)**, the pseudo gradient consists of two components:
>
> $$
> \gamma = 1 - \mathcal{H}(P_n) \quad \text{and} \quad \zeta = P_n - \frac{1}{C},
> $$
>
> where
>
> $$
> P_n = \text{softmax}\left(\frac{1}{n}\sum_{i=1}^{n} P_{cache}(\tilde{\mathbf{f}}_{\mathbf{x}}^{i})\right)
> $$
>
> is the averaged cache prediction based on noise-augmented features and $\mathcal{H}(\cdot)$ denotes the normalized entropy. Here, $\gamma$ controls the **update intensity** (**high-confidence samples are updated more, while low-confidence samples are updated less.**), and $\zeta$ controls the **update direction,** pushing higher probabilities than uniform and pushing down lower probabilities than uniform, which is **exactly what entropy minimization tends to do.**
>
> In cache-based TTA, the prediction for class $c$ is computed via the similarity between the test feature $\mathbf{f}_\mathbf{x}$ and cached features $f_i$ for $i=1,\ldots,N$ in $\mathcal{F}$. For a cached feature $f_c$ belonging to class $c$, the corresponding logit is
>
> $$
> z_c = \mathbf{f}_{\mathbf{x}}^T f_c,
> $$
>
> and $p_c$ is the probability after softmax. The gradient of entropy with respect to $f_c$ can be decomposed as:
>
> $$
> \frac{\partial H}{\partial f_c} = \frac{\partial H}{\partial p_c} \frac{\partial p_c}{\partial z_c} \frac{\partial z_c}{\partial f_c} = \frac{\partial H}{\partial p_c} \frac{\partial p_c}{\partial z_c} \mathbf{f}_{\mathbf{x}},
> $$
>
> where $\frac{\partial z_{c}}{\partial f_{c}} = \mathbf{f_{x}}$ follows from the inner product $z_{c} = \mathbf{f_{x}}^{T} f_{c}$.
>
>
> The first term is
>
> $$
> \frac{\partial H}{\partial p_c} = \frac{\partial}{\partial p_c}\left(-\sum_{k=1}^C p_k \log p_k\right) = -\log p_c - 1,
> $$
>
> serving as the directional component that determines how each probability should be adjusted. The second term stems from the **softmax Jacobian**: for softmax $p_c = \frac{e^{z_c}}{\sum_k e^{z_k}}$, we have
>
> $$
> \frac{\partial p_c}{\partial z_c} = p_c(1 - p_c).
> $$
>
> Critically, since $p_c \in (0,1)$, this term is always **positive** and **only modulates the update magnitude without affecting its direction**. Therefore, the directional behavior of entropy minimization is determined by $\frac{\partial H}{\partial p_c} = -\log p_c - 1$, while the update naturally includes the test feature $\mathbf{f}_{\mathbf{x}}$ as a multiplicative factor.
>
> **Linear approximation via Taylor expansion.**
> The nonlinearity in $-\log p_c - 1$ can be numerically unstable when probabilities approach 0 or 1, especially under distribution shifts. We derive a stable linear approximation by performing **first-order Taylor expansion** around the uniform distribution $p_c = \frac{1}{C}$:
>
> $$
> -\log p_c - 1 \approx -\log \frac{1}{C} - 1 + \frac{d}{dp_c}\left(-\log p_c - 1\right)\bigg|_{p_c=\frac{1}{C}} \left(p_c - \frac{1}{C}\right).
> $$
>
> Computing the derivative:
>
> $$
> \frac{d}{dp_c}(-\log p_c - 1) = -\frac{1}{p_c},
> $$
>
> which evaluated at $p_c = \frac{1}{C}$ gives $-C$. The constant term $-\log \frac{1}{C} - 1 = \log C - 1$ does not affect the gradient direction and can be absorbed into the pseudo learning rate $\eta$. Thus:
>
> $$
> -\log p_c - 1 \approx -C\left(p_c - \frac{1}{C}\right) + \text{const}.
> $$
>
> Combining with the test feature factor, the gradient direction is approximated as
>
> $$
> -C\left(p_c - \frac{1}{C}\right) \cdot \mathbf{f}_{\mathbf{x}}.
> $$
>
> Therefore, **our cache update in Eq. (9):**
>
> $$
> \hat{\mathcal{F}} = \mathcal{F} + \eta \cdot \gamma \cdot \zeta \cdot \mathbf{f}_{\mathbf{x}}
> $$
>
> directly implements this approximation.

---

> ### Author Response · Authors · 2025-11-21
>
> - We also  conducted experiments using true-gradient refinement based on entropy minimization. The results are provided in the following table:
> Across all datasets tested, we found that **the performance of the pseudo-gradient closely matches that of the true entropy-minimization gradient with less computational cost**, empirically demonstrating the proposed pseudo-gradient.
>
> |                 | Aircraft | Caltech101 | Cars  | DTD   | EuroSAT | Flower102 | Food101 | Pets  | SUN397 | UCF101 | ImageNet | ImageNet-A | ImageNet-V2 | ImageNet-R | ImageNet-S | Mean        |
> |-----------------|----------|------------|-------|-------|---------|-----------|---------|-------|--------|--------|----------|------------|-------------|------------|------------|-------------|
> | Pseudo-gradient | 25.86    | 94.76      | 67.06 | 48.00    | 65.64   | 74.17     | 86.20 | 90.95 | 68.00     | 71.34  | 70.54    | 61.22      | 65.83       | 81.81      | 50.82      | 68.15 |
> | True gradient   | 26.73    | 94.93      | 66.92 | 48.78 | 66.01   | 74.25     | 88.13   | 90.46 | 67.43  | 71.23  | 71.11    | 61.11      | 65.90        | 81.33      | 51.32      | 68.38      |
>
> We have added this experiment and discussions in Appendix A.
>
> ---
>
> We are eager to hear your feedback. We’d deeply appreciate it if you could let us know whether your concerns have been addressed.

---

### Official Review · Reviewer_2VNJ · 2025-11-03

**Soundness:** 3
**Presentation:** 3
**Contribution:** 2
**Rating:** 4
**Confidence:** 4

**Summary:**

This paper proposes ROSE-TTA, a cache-based test-time adaptation framework designed to improve the robustness of vision-language models (specifically CLIP) under distribution shifts. The method integrates a noise-aware uncertainty measure for more reliable cache construction, a graph-based structural completion strategy to address class imbalance, and a sample-specific refinement mechanism to enhance local adaptation. Extensive experiments across 15 datasets demonstrate the effectiveness and efficiency of the proposed approach.

**Strengths:**

1. The dual use of global structural completion and local instance-aware refinement shows thoughtful architectural design.
2. Extensive experiments on the cross-dataset and out-of-distribution (OOD) benchmark demonstrate the effectiveness of the proposed method.

**Weaknesses:**

1. To more convincingly demonstrate the effectiveness and robustness of the proposed method, it is recommended to include experiments on standard OOD corruption benchmarks, such as ImageNet-C, CIFAR-10-C, and CIFAR-100-C.
2. In the Implementation Details, the authors state that “The coefficient $\alpha$ is tuned individually for each dataset to adapt to the specific scenario.” It would be helpful to include an ablation study or sensitivity analysis of $\alpha$ (e.g., varying $\alpha$ over a reasonable range) to show how sensitive the performance is to this hyper-parameter.
3. Some descriptions are unclear. For example, in Eqn. (3), it is not specified how $\sigma$ is defined or estimated. Please clarify the exact meaning of $\sigma$.
4. In Lines 240–241, the authors claim that the introduced textual embedding “is leveraged to complete the insufficient class information and mitigate class imbalance.” It remains unclear why or how incorporating textual embeddings complements class information and alleviates class imbalance. Providing a more detailed explanation or additional empirical evidence (e.g., analysis or ablations) would make this argument more convincing.

**Questions:**

Please refer to the Weaknesses.

---

> ### Author Response · Authors · 2025-11-21
>
> Thank you for your thoughtful review and valuable questions! Below, we address your questions and indicate the changes we’ve made thanks to your suggestion.
>
> > **W1:To more convincingly demonstrate the effectiveness and robustness of the proposed method, it is recommended to include experiments on standard OOD corruption benchmarks, such as ImageNet-C, CIFAR-10-C, and CIFAR-100-C.**
>
> **Response:** Thank you for your kind suggestions! We **add experiments on OOD corruption benchmarks including ImageNet-C, CIFAR-10-C, and CIFAR-100-C** using CLIP-ViT-B/16 as our backbone in Appendix C.1. We compared our proposed method with zero-shot CLIP, TDA and **BCA (the most competitive baseline)** as shown below.
>
>
>
> |  | Guassian | Shot | Impulse | Defocus | Glass | Motion | Zoom | Snow | Frost | Fog | Brightness | Contrast | Elastic | Pixelate | JPEG | Mean |
> |--------|-----------|------|---------|---------|-------|--------|------|------|-------|-----|-----------|---------|---------|----------|------|------|
> | **ImageNet-C** |
> | CLIP | 18.56 | 12.11 | 19.45 | 31.22 | 23.12 | 32.15 | 26.25 | 34.59 | 34.17 | 45.35 | 57.11 | 30.94 | 48.93 | 45.01 | 35.14 | 32.94 |
> | TDA | 19.54 | 13.46 | 19.9 | 32.76 | 26.35 | 34.32 | 27.79 | 36.42 | 35.47 | 47.79 | 58.16 | 31.92 | 49.22 | 45.05 | 35.8 | 34.26 |
> | BCA | 19.76 | 13.44 | 20.38 | 33.08 | 27.84 | 34.62 | 27.88 | 38.12 | 35.96 | 52.26 | 59.02 | 32.38 | 49.20 | 45.08 | 35.94 | 34.99 |
> | Ours | **20.18** | **13.57** | **21.56** | **33.67** | **29.78** | **35.19** | **28.12** | **40.17** | **36.66** | **60.89** | **59.94** | **33.26** | **49.38** | **45.18** | **36.21** | **36.25** |
> | **CIFAR-10C** |
> | CLIP| 46.22 | 50.17 | 45.23 | 65.45 | 43.33 | 60.14 | 68.54 | 69.12 | 71.11 | 55.23 | 80.45 | 44.53 | 55.25 | 52.17 | 60.31 | 57.82 |
> | TDA | 48.66 | 53.24 | 46.26 | 67.09 | 43.55 | 63.17 | 69.97 | 70.02 | 72.57 | 56.13 | 81.90 | 45.59 | 56.84 | 53.71 | 60.91 | 59.31|
> | BCA | 49.08 | 53.72 | 46.18 | 68.00 | 44.55 | 65.92 | 70.82 | 71.38 | 72.98 | 56.75 | 81.98 | 47.05 | 56.35 | 54.02 | 61.52 | 60.03 |
> | Ours | **50.14** | **54.76** | **46.31** | **69.73** | **46.17** | **69.56** | **72.45** | **73.52** | **74.23** |**57.86** | **82.04** | **49.23** | **56.97** | **54.75** | **62.69** | **61.36**|
> | **CIFAR-100C**  |
> | CLIP | 28.15 | 28.18 | 20.17 | 38.22 | 20.85 | 35.24 | 42.56 | 40.14 | 42.88 | 30.13 | 52.19 | 20.89 | 29.66 | 24.15 | 33.11 | 32.43 |
> | TDA | 29.18 | 29.84 | 24.68 | 39.44 | 20.99 | 37.44 | 43.84 |42.86 | 44.24 | 30.72 | 53.72 | 22.85 | 30.78 | 25.13 | 33.81 | 33.97 |
> | BCA | 29.66 | 30.10 | 25.96 | 40.36 | 21.64 | 37.70 | 43.62 | 42.92 | 46.28 | 30.79 | 54.90 | 23.74 | 32.16 | 25.18 | 33.88 | 34.51 |
> | Ours | **30.22** | **30.46** | **27.89** | **41.58** | **22.45** | **38.11** | **44.19** | **43.00** | **48.69**| **30.87** | **56.24** | **24.78** | **33.94** | **25.22** | **33.96** | **35.44** |
>
> We found **our ROSE-TTA stably outperforms CLIP, TDA and BCA on all three OOD corruption benchmarks**, demonstrating the effectiveness and robustness of our method.
> We added this experiment and discussions in Appendix C.1.

---

> ### Author Response · Authors · 2025-11-21
>
> > **W2: It would be helpful to include an ablation study or sensitivity analysis of  $\alpha$ (e.g., varying  α over a reasonable range) to show how sensitive the performance is to this hyper-parameter.**
>
> **Response:** Thank you for pointing out this issue! In response, we added experiments across all datasets to evaluate the performance under different values of $\alpha$ in Appendix C.3.
>
> | Dataset       | Aircraft | Caltech101 | Cars  | DTD   | EuroSAT | Flower102 | Food101 | Pets   | SUN397 | UCF101 | ImageNet | ImageNet-A | ImageNet-V2 | ImageNet-R | ImageNet-S | Mean   |
> |---------------|----------|------------|-------|-------|---------|-----------|---------|--------|--------|--------|----------|------------|-------------|------------|------------|--------|
> | $\alpha$ = 0  | 24.70    | 94.16      | 65.80 | 44.50 | 47.67   | 71.42     | 86.10   | 89.15  | 66.62  | 66.77  | 70.04    | 59.43      | 64.35       | 80.33      | 49.40      | 65.36  |
> | BCA | **28.59**    | 94.69      | 66.86 | 53.49 | 56.63   | 73.12     | 85.97 | 90.43  | **68.41** | 67.59  | 70.22    | 61.14     | 64.90   | 80.72      |**50.87**     | 67.58 |
> | $\alpha$ = 1  | 24.77    | 94.45      | 66.92 | 47.78 | 60.12   | 73.42     | 86.12   | **90.95** | 67.43  | **71.34** | 70.11    | 60.11      | 64.90       | 81.33      | 50.32      | 67.40  |
> | $\alpha$ = 3  | 25.86| 94.50      | **67.06** | **48.00** | **65.64** | 73.67     | 86.15   | 90.42  | 68.00 | 70.33  | **70.54** | **61.22**  | 65.68       | 81.56      | 50.45      | **67.94**  |
> | $\alpha$ = 5  | 25.23    | **94.76**  | 66.77 | 47.90 | 65.50   | **74.17** | 86.16   | 90.56  | 67.56  | 69.57  | 70.33    | 60.58      | 65.77       | **81.81**  | 50.82  | 67.83  |
> | $\alpha$ = 10 | 25.69    | 94.68      | 67.02 | 47.22 | 63.25   | 74.11     | **86.16** | 90.66  | 67.78  | 70.12  | 70.42    | 61.01      | **65.83**   | 81.23      | 50.64      | 67.73 |
>
> Our results show that **(i) our method has relatively stable performance with changes in $\alpha$**, **(ii) our method significantly outperforms the degenerated case (using only zero-shot CLIP predictions) where $\alpha=0$**, and **(iii) across the majority of $\alpha$ values tested, our method consistently outperforms the most competitive baseline BCA**, demonstrating that even without careful $\alpha$ search, our approach maintains its effectiveness. This robustness validates the effectiveness  of our cache-based refinement strategy.
> > **W3: Some descriptions are unclear. For example, in Eqn. (3), it is not specified how  $\sigma$  is defined or estimated. Please clarify the exact meaning of $\sigma$.**
>
> **Response:** Thank you for your comments. We **added the following details in Sec 3.2 and conduct a corresponding sensitivity analysis in Appnedix C.4.**
>
> - **Definition and Setting of $\sigma$:** In our implementation, **$\sigma$ is a scalar hyperparameter that controls the magnitude of Gaussian noise applied to the test features**. We empirically set $\sigma$ = 0.1 based on preliminary experiments that balance perturbation strength and feature semantic preservation (see below). This value is kept constant throughout the adaptation process.
>
>
> - **Empirical Justification on $\sigma$ Selection:**
> We conducted sensitivity analysis on $\sigma$ ∈ {0, 0.05, 0.1, 0.15, 0.2} across all datasets, where $\sigma$ = 0 means "Entropy only". **Performance is relatively the best around σ = 0.1**. If σ is too small, The perturbations become negligible, failing to meaningfully test the sample's robustness. If σ is too large, excessive noise can corrupt the semantic information in the features, causing even truly reliable samples to exhibit unstable predictions.  This validates our choice of σ = 0.1 as a robust default.
>
> | $\sigma$ | Cross Datset Avg | OOD Avg |
> |-------|------------------|---------|
> |     0 |            68.21 |   65.13 |
> |  0.05 |            69.17 |   66.01 |
> |   0.1 |             **69.20** |   **66.04** |
> |  0.15 |            69.11 |   65.98 |
> |   0.2 |            68.88 |   65.56 |

---

> ### Author Response · Authors · 2025-11-21
>
> > **W4: The paper does not clearly explain how textual embeddings complement class information and mitigate class imbalance, providing a more detailed explanation or additional empirical evidence (e.g., analysis or ablations) would make this argument more convincing.**
>
> **Response:** Sorry for the unclear statement. We intend to say **incorporating the textual embeddings helps to alleviate the biased cache logits caused by the class imbalance, rather than directly mitigating the class imbalance problem.**
>
> **Class imbalance problem in our setting:**
> - During the early adaptation stage, random arrival leads to **unequal cache capacities across classes**: some classes may have accumulated 5 cached samples (full capacity), while others may have only 1-2 samples, or even no samples at all. This causes **biased predictive logits—classes with more cached samples contribute dominant logits to the final prediction**, while classes with few or no cached samples contribute negligible or zero logits.
>
> **More explanation on textual embeddings alleviate biased logits:**
> By incorporating textual embeddings via the textual graph, we **reconstruct the imbalanced cache graph** to provide complementary information for underrepresented classes. Specifically, the textual graph leverages semantic relationships between classes to **enhance the logits of categories with insufficient cache samples**, thereby producing more balanced predictions. The ablation study in Table 3 of the main paper demonstrates the effectiveness of this component.
>
> **Addtional empirical evidence:**
> Here, we further provide experiments of  **predictive cache logits with and without the textual graph** across different cache capacities **at 10% testing progress** to show alleviation of the biased logits. We average  all cache logits of categories with each cache capacity.
>
> | Dataset | | Cache Capacity=0 | Cache Capacity=1 | Cache Capacity=2 | Cache Capacity=3 | Cache Capacity=4 | Cache Capacity=5 |
> |---------|-------------|-------------|-------------|-------------|-------------|-------------|-------------|
> | Imagenet | w/o textual graph | 0.0000 | 0.0006 | 0.0007 | 0.0008 | 0.0009 | 0.0011 |
> | | w/ textual graph | 0.0007 | 0.0008 | 0.0009 | 0.0009 | 0.0010 | 0.0010 |
> | Caltech101 | w/o textual graph | 0.0000 | 0.0081 | 0.0090 | 0.0103 | 0.0118 | 0.0221 |
> | | w/ textual graph | 0.0064 | 0.0093 | 0.0095 | 0.0101 | 0.0109 | 0.0142 |
> | oxford_flowers | w/o textual graph | 0.0000 | 0.0060 | 0.0074 | 0.0106 | 0.0154 | 0.0211 |
> | | w/ textual graph | 0.0073 | 0.0082 | 0.0095 | 0.0096 | 0.0101 | 0.0132 |
> | stanford_cars | w/o textual graph | 0.0000 | 0.0029 | 0.0042 | 0.0039 | 0.0046 | 0.0054 |
> | | w/ textual graph | 0.0039 | 0.0041 | 0.0045 | 0.0044 | 0.0049 | 0.0052 |
>
>  Without the textual graph, classes with no cached samples  contribute **zero logits**, making them completely invisible during prediction. With the textual graph, **even classes with zero cached samples receive non-zero logits** (e.g., 0.0007 for ImageNet, 0.0064 for Caltech101), enabling them to contribute to the prediction. Moreover, the logits across different cache capacities are more balanced.
>    This demonstrates that the textual graph effectively **alleviates the bias caused by unequal cache capacities**, ensuring that underrepresented classes still participate meaningfully in predictions during the early adaptation stage.We added this experiment and discussions in Appendix C.8.
>
> ---
>
> We hope the above revisions and explanations resolve your concerns, and we remain at your disposal for any further discussion.

---

### Author Response · Authors · 2025-11-23
**General responses and manuscript revision summary**

Dear reviewers and AC,

We sincerely thank all reviewers and AC for their great effort and constructive comments on our manuscript. We appreciate this opportunity to engage in discussions with the reviewers. During the rebuttal period, we have been focusing on these beneficial suggestions from the reviewers and doing our best to add several experiments and revise our manuscript. We believe our current carefully revised manuscript can address all the reviewers' concerns.

As reviewers highlighted, we believe our paper  **unify framework that addresses both cache construction and utilization reliability to tackle  cache-based test-time adaptation for vision-language models** (Reviewer 2VNJ, Reviewer rmg6, Reviewer PpY7, Reviewer 7oFc). We also appreciate that the reviewers found the proposed method **novel and well-motivated with multiple new techniques** (Reviewer 2VNJ, Reviewer rmg6, Reviewer 7oFc), **thoughtfully designed by combining**  **noise-aware cache selection** (Reviewer PpY7) **, graph-based completion, and sample-specific refinement** (Reviewer 2VNJ, Reviewer rmg6), **training-free and  efficient for deployment** (Reviewer rmg6), as well as **comprehensive ablations demonstrating effectiveness** (Reviewer 2VNJ, Reviewer rmg6).

Moreover, we thank the reviewers for pointing out the need for **clarifying the novelty and significance of our contributions** (Reviewer PpY7), **more extensive validation on consistency and universality of improvements** (Reviewer 2VNJ, Reviewer rmg6, Reviewer PpY7), **hyperparameter sensitivity analysis** (Reviewer 2VNJ, Reviewer rmg6), **comprehensive efficiency and overhead analysis** (Reviewer rmg6, Reviewer PpY7, Reviewer 7oFc), **cache accuracy and dynamics comparisons** (Reviewer PpY7, Reviewer 7oFc),  **noise augmentation baseline comparisons** (Reviewer PpY7), **theoretical justification for pseudo-gradients** (Reviewer rmg6), **robustness to sample ordering** (Reviewer 7oFc), **clearer explanations of technical details** (Reviewer 2VNJ), **progress curves versus TDA**(Reviewer PpY7) as well as **clarification on presentation issues** (Reviewer PpY7, Reviewer 7oFc). In response to these comments, we have carefully revised and enhanced our manuscript with the following important changes and added experiments:

* **[Reviewer PpY7]** We **add clarification** highlighting ROSE-TTA as the first unified framework co-designing cache lifecycle, introducing noise-enhanced stability metric, graph-based completion and pioneering pseudo-gradient formulation.

* **[Reviewer 2VNJ, Reviewer rmg6, Reviewer PpY7]** We **add OOD corruption experiments** on ImageNet-C, CIFAR-10-C, and CIFAR-100-C in Appendix C.1, demonstrating consistent improvements across 15 corruption types.

* **[Reviewer 2VNJ, Reviewer rmg6]** We **add α sensitivity analysis** across all datasets in Appendix C.3, showing stable performance, demonstrating effectiveness even without careful hyperparameter search.

* **[Reviewer rmg6, Reviewer PpY7, Reviewer 7oFc]** We **add efficiency benchmarks** in Section 5.2 with **component-level breakdown** in Appendix B.

* **[Reviewer PpY7, Reviewer 7oFc]** We **add cache dynamics analysis** in Appendix C.6, **ablation on eviction strategies** in Appendix C.7, and **cache accuracy comparisons** in Figure 7 (Appendix C.2) similar to Figure (a) and (b).

* **[Reviewer PpY7]** We **add scalability analysis** in Appendix B.2 and **n=0 baseline** in Appendix C.5.

* **[Reviewer rmg6, Reviewer PpY7]** We **add BCA to progress comparison curves** in the revised Figure 5, demonstrating our method's advantages during early adaptation stages compared to both TDA and BCA.

* **[Reviewer 7oFc]** We **add robustness analysis under random sample orderings**  in Figure 6 (Section 5.2), showing stable performance and **clarify Equation (5)**.

* **[Reviewer rmg6]** We **add theoretical analysis** deriving the connection between pseudo and gradient  with **empirical comparisons**.

* **[Reviewer 2VNJ]** We **add detailed definition and empirical justification for the noise magnitude σ** in Section 3.2 with sensitivity analysis  in Appendix C.4.

* **[Reviewer 2VNJ]** We **add empirical analysis of predictive cache logits** with and without textual graph  Appendix C.8, demonstrating how textual embeddings alleviate biased predictions.

* **[Reviewer PpY7]** We **revise Table 3 to report averages separately** for cross-dataset and OOD benchmarks. And **fix terminology inconsistency** from ROSE-TDA to ROSE-TTA.

These updates are temporarily highlighted in $\color{blue}{\text{\textit{blue}}}$  for facilitating checking.

We hope our response and revision could address all the reviewers' concerns, and are more than eager to have further discussions with the reviewers in response to these revisions.

Many thanks,

Submission17055 Authors

---

### Comment · Area_Chair_4k5S · 2025-11-26
**Reminder: Discussion Phase Engagement Needed**

Dear Reviewer 2VNJ, rmg6 and PpY7:

As the deadline for the discussion phase is approaching in less than one week, could you kindly engage in the discussion with the other reviewers and provide your response to the authors’ rebuttal?

Best regards,

AC

---

### Note · Program_Chairs · 2026-01-17
**Submission Desk Rejected by Program Chairs**

The following references in this submission do not refer to real documents and/or have major errors in bibliographic information:

 Zhi-Fan Feng, Jie Zhou, Wen-Huang Li, and Zhen-Hua Zhang. Difftpt: Leveraging diffusion models for test-time prompt tuning. In arXiv preprint arXiv:2305.13998, 2023.

Sarel Karmanov, Leonid Karlinsky, Shir Doveh, Assaf Arbelle, Rogerio Feris, Raja Giryes, and Alex Bronstein. Promptalign: Bridging the gap between model and human preferences via natural language feedback. arXiv preprint arXiv:2312.01459, 2023.

Jingyi Tang, Jiaxing Wang, Jingdong Yang, Jingyao Liu, and Hongdong Li. Histpt: Historical test-time prompt tuning for vision foundation models. In arXiv preprint arXiv:2312.09051, 2023.